# Last Interglacial sea-level history from speleothems: a global standardized database

Oana A. Dumitru[1], Victor J. Polyak[2], Yemane Asmerom[2], Bogdan P. Onac[3]

[1]Biology and Paleo Environment Department, Lamont-Doherty Earth Observatory, Columbia University, Palisades, NY 10964, USA

[2]Department of Earth and Planetary Sciences, University of New Mexico, Albuquerque, NM 87131, USA

[3]School of Geosciences, University of South Florida, 4202 E. Fowler Ave., NES 107, Tampa, FL 33620, USA

*Correspondence to*: Oana A. Dumitru (odumitru@ldeo.columbia.edu)

**Abstract.** Speleothems (secondary cave carbonate precipitates) are archives of valuable information for reconstructing past sea levels as they are generally protected from weathering and erosion by their location and can be dated with U-series methods. Two main categories of speleothems are recognized as sea-level indicators: phreatic overgrowth on speleothems (POS) and submerged vadose speleothems (SVS). POS have the great advantage that they precipitate on preexisting supports (vadose speleothems or cave walls) at a brackish water level equivalent to sea level when air-filled chambers of coastal caves are flooded by rising sea. SVS are also useful, but sea level is inferred indirectly as periods of growth provide constraints on maximum sea-level positions, whereas growth hiatuses, sometimes difficult to observe, may indicate times when cave passages are submerged by sea highstands, hence they record minimum sea-level elevations.

Here we describe a compilation that summarizes the current knowledge of the complete last interglacial (in its the broadest sense from ~140 to 70 ka, also known as marine isotope stage (MIS) 5) sea level captured by speleothems. We used the framework of the World Atlas of Last Interglacial Shorelines (WALIS), a comprehensive sea-level database, to provide a standardized format in order to facilitate scientific research on MIS 5 sea level. The discussion is focused on MIS 5e, but records that capture MIS 5d, 5c, 5b, and 5a are also included. We present the data from 71 speleothems (36 sea-level index points and 37 limiting points) in coastal caves located in ten different locations and we include the spatial coverage, the samples used and their accuracy as indicators of sea level, U and Th isotopes used to generate the chronologies, and their scientific relevance to understand past sea-level changes. Furthermore, the paper emphasizes the usefulness of these indicators not only to render information regarding the eustatic sea level, but also for their contribution to refine the glacial isostatic adjustments models and to constrain regional tectonic uplift rates. The standardized sea-level database presented here is the first of its kind derived from speleothems and contains all the information needed to assess paleo relative sea levels and the chronological constraints associated with them. The database is available open-access at http://doi.org/10.5281/zenodo.4313860 (Dumitru et

al., 2020). We refer the readers to the official documentation of the WALIS database at: https://walis-help.readthedocs.io/en/latest/ where the meaning of each field is explained in detail.

## 1 Introduction

Understanding sea-level changes during the last interglacial period (MIS 5e; in its strictest sense from 130–116 ka) is key to assess the behavior of ice sheets in a warmer world as MIS 5e is considered a potential analog for the future sea-level rise due to anthropogenic global warming since temperatures were on average ~ 1.5 °C higher than today (relative to the AD 1961–1990 period; Turney and Jones, 2010). Sea level indicators formed during MIS 5e are often better preserved compared to those from earlier interglacial periods, and thus, relative sea level (RSL) during this time interval is especially informative (Capron et al., 2019). However, significant uncertainties regarding the precise timing, duration, and amplitude of MIS 5e sea level remain. The main limiting factor is finding a sea-level indicator that can robustly constrain both water depth and age. Fossil corals can be dated to relatively high precision but have meter scale uncertainties in the reconstructed sea level (Hibbert et al., 2016; Chutcharavan and Dutton, 2021). Other indicators, such as erosional notches (Bini et al., 2014; Antonioli et al., 2015) or flank margin caves (Carew and Mylroie, 1995; Mylroie et al., 2020) pinpoint sea level, but lack tight age control. For this reason, there is a growing demand in exploring sea-level indicators that can simultaneously provide a robust chronology and have a clear indicative meaning.

Relevant sea-level markers such as flank margin caves (Mylroie et al., 2020), tidal notches (Bini et al., 2014), phreatic overgrowths on speleothems, and vadose submerged speleothems (Richards et al., 1994; Onac et al., 2012) are unique to coastal karst environments (Van Hengstum et al., 2015). Over the past decade, there has been a growing interest in cave deposits that allow for sea level reconstructions, which include: phreatic overgrowths on speleothems (POS; indicative of the position of sea-level stillstands) and submerged vadose speleothems (SVS; providing maximum elevations of sea-level position). A very large number of speleothem records have been reported and a comprehensive compilation by the Speleothem Isotopes Synthesis and AnaLysis working group was used for multiple climate reconstructions and model evaluations (Atsawawaranunt et al., 2018; Comas-Bru et al., 2020). However, the majority of these studies are mainly directed towards paleoclimate reconstructions and only a handful focused on documenting sea-level changes as recorded by these deposits. The idea of using speleothems in reconstructing Quaternary sea-level changes dates back five decades (Benjamin, 1970; Spalding and Mathews, 1972; Ginés and Ginés, 1974). Relative to corals, an advantage of employing speleothems as sea-level markers is that the dense cave calcite is less susceptible to alteration. An additional benefit is that karst caves provide an excellent and sheltered environment in which these deposits are well preserved and protected against processes that disrupt or destroy other terrestrial archives.

## 1.1 The relationship between POS/SVS deposition and sea-level changes

Phreatic overgrowths on speleothems. POS form on submerged cave walls and pre-existing vadose speleothems at and just below the water table (Ginés et al., 2012), when seawater mixes with meteoric water inside caves that are located in the close proximity to the coastline (within 300 m). The pre-existing vadose speleothems become partly submerged in the resulting brackish water (Fig. 1a). Previous petrographic investigations of these deposits suggested that the major control on carbonate precipitation is the ability of $CO_2$ to degas across the water-air interface (Pomar et al., 1976; Csoma et al., 2006). These findings are supported by present-day observations that indicate the upper 40 cm of the water column being supersaturated with respect to calcium carbonate allowing for POS to form (Boop et al., 2014). While meteoric-marine mixing zones are mostly referred as sites of extensive dissolution, aragonite or calcite precipitation occurs when a high concentration gradient between $pCO_2$ of the cave water and atmosphere exists. Therefore, faster degassing is expected to happen in caves with low $CO_2$ partial pressure ($pCO_2$) in their atmosphere. Corrosion of carbonate minerals was noticed in some Mallorcan caves, particularly when approaching the halocline; however, both calcite and aragonite are presently precipitating at the water table in the mixing zone, where numerical model predicts dissolution (Csoma et al., 2006).

With very few exceptions, the morphology of the POS is clearly different from that of speleothems precipitated at the fresh water level in pools from non-coastal caves, e.g., shelfstones and subaqueous freshwater pool spar, on which the overgrowths are truncated in the upper part and mainly accrete under the water level. Furthermore, the carbonate deposition of these speleothems is not symmetric with respect to the water level and the tide range, which is a particularity of POS. The only instance when the shape of POS is asymmetric (i.e., form only under the water level) is when the preexisting vadose speleothem (i.e., stalagmite) was not long enough to capture the full range of the tide, which is responsible for their spherical or elliptical morphology. POS can take a variety of shapes and sizes, (Fig. 1e, f), depending on the morphology of the vadose support, for how long they were immersed in the cave's brackish water, and the tide amplitude. Only few petrological and geochemical studies have been performed so far (Pomar et al., 1976; Ginés et al., 2005; 2012; Csoma et al., 2006). The mineralogical and crystallographical data indicate calcite as the dominant phase with fibrous, elongated, and isometric crystals, but radial-fibrous/acicular aragonitic fabric can exceed 70% in some samples (Ginés et al., 2012). A limited number of stable isotopes analyses showed an isotopic evolution towards heavier composition through the MIS 5e and 5a possibly due to excessive marine water intrusion in the cave ponds (Vesica et al., 2000). More in-depth studies have been undertaken to investigate the POS precipitation conditions and the relationships between surface conditions (temperature, barometric pressure, precipitation, tidal level of the sea) and the microenvironment of coastal caves (temperature, $pCO_2$, and water level; Boop et al., 2014). The distinction between POS and shelfstones (flat deposits attached to cave walls or on partly immersed speleothems that grow inwards from the edge of the pool/speleothem) is clearly described in Onac et al. (2012).

The hydraulic gradient between caves hosting POS and the sea is insignificant, since the caves are proximal to the coastline and thus, the brackish water table in these caves is, and was in the past, coincident with sea level. As long as sea level remains

at the same elevation, carbonate precipitation occurs within the tidal range at the air-water interface (Vesica et al., 2000; Fornós et al., 2002; Dorale et al., 2010; Polyak et al., 2018). Therefore, the presence of POS horizons at different elevations precisely marks the positions of paleo-water tables and consequently their associated sea-level position. Given the precipitation mechanism (Fig. 1d), POS arguably provide the most precise and less ambiguous indicator of the absolute elevation of sea-level position. POS are meaningful *sea-level index points* because they provide spatial geographic positioning, accurate elevation, and absolute ages. This type of cave deposit has only been identified in a few places worldwide: Mallorca (Ginés and Ginés, 1974; Vesica et al., 2000; Tuccimei et al., 2007; Dorale et al., 2010; Polyak et al., 2018; Dumitru et al., 2019), Sardinia (Tuccimei et al., 2012), Nansei islands, Japan (Pacific Ocean; Urushibara-Yoshino, 2003; Minami Daito Island; Miklavič et al., 2018), Christmas Island, Australia (Indian Ocean; Grimes, 2001), and Mexico (Jenson, 2018). Similar deposits have been recently described and dated from Santa Catalina Cave in Cuba (De Waele et al., 2017; 2018). While of different morphologies, such as mushroom caps and bench-like encrustations similar to cave shelfstones (called balconies) they appear to be POS (Fig. 1f). Earlier investigations of these speleothems suggested that their formation is highly dependent on microbial activity and water level fluctuation (Bontognali et al., 2016).

Submerged vadose speleothems. Speleothems such as stalactites and stalagmites form in air-filled passages (Fig. 1a), thus periods of their growth indicate times when sea level was lower; hence they are *sea-level terrestrial limiting points*. For vadose speleothems that are subject to sea-level submergence, hiatuses (i.e., no carbonate deposition) can be correlated with periods when sea level rose and inundated the cave causing speleothem growth to cease. Important to note is that a prolonged pause in speleothem growth is not always caused by sea-level rise; changes in hydrology and hydrochemistry above the cave, for example, undersaturation of drip water or cessation of dripping (drought or blockage of drainage path), can also lead to interruption of speleothem deposition (Onac et al., 2012). When growth cessation is sea-level related, particular mineralogical and/or biological features can be visible using petrography. Some of these include: i) corroded layers when dissolution happens at the halocline; ii) biogenic encrustations (Fig. 1b, c); iii) traces of marine borings; and iv) deposition of various trace elements or minerals (halite, gypsum, etc.). Details regarding ways of deciphering different types of growth hiatuses are presented by Onac et al. (2012) and van Hengstum et al. (2015). Dating the carbonate layer immediately above each of these hiatuses provides a minimum estimate of when the cave became air-filled again constraining the minimum age for the sea-level fall. The carbonate layer below a hiatus indicates the maximum age, assuming no post-depositional alteration of the exposed surfaces, for when this location in the cave was air-filled and the sea level was clearly below the speleothem elevation. It is worth noting that the earliest layers deposited above the hiatuses are protected by further carbonate precipitation, whereas those below the hiatuses are susceptible to diagenetic alteration or even dissolution. Thus, such carbonate accumulations provide precise age constrain on the initiation of growth, but may not tell exactly when sea level dropped.

SVS stop growing when sea level is above their elevation and restart their deposition when sea level falls below them (Fig. 1b), thus, they record past sea-level fluctuations indicating when the cave was air-filled or invaded by seawater (Richards et

al., 1994; Moseley et al., 2013). Once the sea level falls again, if conditions are suitable, the speleothem may resume its deposition. Hence, the vadose speleothems can only provide the minimum and maximum age when a particular part of the cave became flooded or air-filled, not precisely when and where the water level was actually located throughout the bulk of the rise-fall cycle (Richards et al., 1994; Surić et al., 2009). Therefore, the growth of vadose speleothems is a limiting point and thus, its relationship with the sea level position must be interpreted correspondingly. The submerged speleothem indicators can be refined by dating alternating continental and marine biogenic overgrowths (serpulid colonies) if they exist and are well preserved (Antonioli et al., 2004; Dutton et al., 2009), but since none of them captured the MIS 5 sea-level stand, they are not discussed in this paper. The age of growth initiation and cessation is dependent on sample position and growth rate. A highly resolved U-Th chronology defines the degree of continuous growth and a high-quality petrographic examination of the sample would support that. These ages can be used to calculate the growth rate, allowing to better define the onset and cessation of deposition for either POS or SVS samples. This information bears significance since one can use the growth rate to project the onset of a hiatus, which in coastal caves provides evidence for when sea level emerged above that particular speleothem elevation. To provide robust chronologies, we refer the readers to the workflow to treat records with hiatuses developed by Comas-Bru et al. (2020).

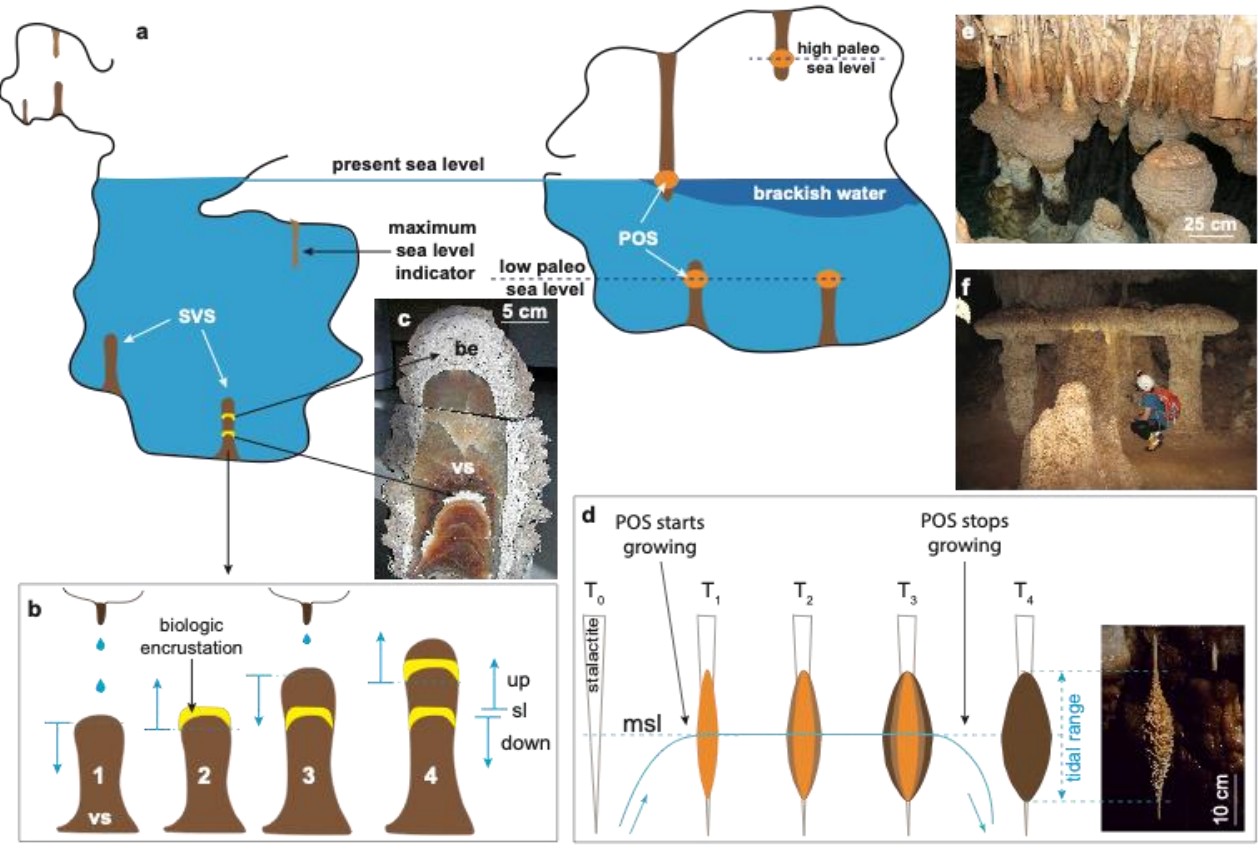

Figure 1 a) Composite diagram showing how submerged vadose speleothems (on the left) and phreatic overgrowths on speleothems (on the right) in littoral caves act as sea-level indicators. b) Conceptual model showing that: b1) growth of SVS indicate times when sea level was lower than their elevation; b2) Hiatus - period of SVS submergence suggesting sea level higher than the SVS elevation; b3) SVS resume its deposition when sea level fell below its elevation. c) SVS (be=biogenic encrustation; vs= vadose speleothem) from Argentarola Cave, Italy (photo courtesy F. Antonioli). d) Conceptual models showing how POS form; as long as sea level remains at the same elevation (T1-T3) POS will precipitate at sea-level and within tidal range and will continue to grow until the sea level drops below the speleothem (T4). e) Late Holocene POS in Cala Varques Cave, Mallorca Island. f) Mushrooms-shaped POS in Santa Catalina Cave, Cuba (photo courtesy B. P. Onac).

## 1.2 Existing MIS 5e sea-level databases and context of our work

Extensive reviews of MIS 5e sea-level indicators (coral reef and marine terraces, shore platforms, beach deposits and ridges, tidal notches, and sea caves) at global scale have been compiled by Kopp et al. (2009), Dutton and Lambeck, (2012), Pedoja et al. (2014), and Hibbert et al. (2016). A step forward was taken by Rovere et al. (2016a) who set the basis for a standardized approach to MIS 5e paleo sea-level reconstructions and interpreted the indicators in terms of the entire geological or

sedimentary facies, rather than considering each of them separately. Standardized sea-level databases allow for regional to global comparisons of records from disparate locations. In turn, this provides a means to disentangle spatial patterns and rates of sea-level change at different timescales. These curated sets of data will ultimately enhance our understanding of the mechanisms driving sea-level fluctuations, thus improving both physical models and statistical reconstructions (Khan et al., 2019).

Speleothems have received little or no attention in prior compilations. A thorough review of the results obtained from the study of Mediterranean submerged speleothems since 1978 and their use to reconstruct past sea level variations has been recently provided by Antonioli et al. (2021). Still, a dedicated cave deposits database is not currently available to the sea-level community. In this context, the present dataset paper aims to compile existing results on cave deposits-derived sea level during the last interglacial period and make them more accessible to paleoclimatology and oceanography community, with the ultimate goal to facilitate research on MIS 5 sea level. This work gathers data from previously published studies, each of which describes the samples analyzed, the isotopic ratios and concentrations used to generate the chronology, and the scientific relevance for interpreting past sea-level changes. Section 2 presents the data, including the criteria for the inclusion of each record such as spatial coverage, the elevation measurements and their uncertainties, and the U-Th methods for their absolute chronology. Section 3 discusses the interpretation of these records and highlights the valuable information they provide for eustatic sea level and the crucial inputs in assessing glacial isostatic adjustment (GIA) models and regional tectonic activity.

## 2 Data description

The data presented here is part of the World Atlas of Last Interglacial Shorelines (WALIS, https://warmcoasts.eu/world-atlas.html), a sea-level database interface developed under the framework of the European Research Council Starting Grant "WARMCOASTS" (ERC-StG-802414), in collaboration with PALSEA (PALeo constraints on SEA level; a PAGES-INQUA working group). WALIS provides a new standardized database and aims to be the most comprehensive compilation of globally distributed (new and old) data on MIS 5 sea-level indicators. The interface allows a large range of data and metadata on relative sea-level indicators and associated ages to be inserted into a mySQL database. An export tool allows downloading the data inserted by the logged user as a multi-sheet Microsoft Excel .xls file. This archive is available open-access as Dumitru et al. (2020; http://doi.org/10.5281/zenodo.4313860). We refer the readers to the official documentation of the WALIS database at: https://walis-help.readthedocs.io/en/latest/, where the meaning of each field is explained in detail. These files will be available in WALIS v1.0, which will provide a user-friendly interface for quick visualization, extraction and downloading of the data. We summarize below the major features of the records which comprise the database in metadata fields that enable easy reuse of the time-series data. In order to ensure high quality data intended for scientific reuse, only results published in peer-reviewed literature were considered.

**2.1 Criteria for records inclusion.** The use of speleothems as sea-level indicators have been reported from several places around the world, however, in this data paper we only present the results that capture sea level during the complete last interglacial (MIS 5; 140–70 ka). The discussion is mostly centered around MIS 5e, but also includes results on the sea-level position during MIS 5d, 5c, 5b, and 5a. A number of POS older or younger than MIS 5 have been published from caves in Mallorca (Vesica et al., 2000; Dumitru et al., 2019; 2021), Japan (Miklavič et al., 2018), and Mexico (Jenson, 2018), but they are not included in this paper. Similarly, SVS (some of which at much lower elevations than the ones presented here) have been identified in several submerged caves in the Bahamas and Mediterranean, but their ages are far too old (Richards, 1995; Smart et al., 2008) or young (Spalding and Mathews, 1972; Beck et al., 2001; Hoffmann et al., 2010; Arienzo et al., 2015; 2017; Antonioli et al., 2021) to be directly relevant for MIS 5 sea-level reconstruction.

This database includes a total of 71 speleothems (36 sea-level index points and 37 limiting points) of variable quality. All the POS records have precisely measured elevations, narrow indicative range, and decimetric RSL uncertainties, hence they are excellent sea-level indicators. A guide for SVS records' evaluation as terrestrial limiting points can be found in Table 1 of WALIS' official documentation (https://walis-help.readthedocs.io/en/latest/Relative%20Sea%20Level/). Worth noting is that the ages that are near but outside the MIS 5 range are not included in the discussion of this paper, but, for completeness, they are entered in the database.

**Table 1. Sites included in data product. Location and name of the caves, type of speleothems, and the corresponding references.**

| Location | Cave name | Type of speleothem | Reference |
|---|---|---|---|
| Mallorca, Spain | Cova de Cala Varques A | POS | Dorale et al., 2010; Polyak et al., 2018 |
| | Cova de Cala Varques B | | Polyak et al., 2018 |
| | Cova del Dimoni | | |
| | Cova de Cala Falcó | | |
| | Cova des Pas de Vallgornera | | |
| | Cova Genovesa | | |
| | Cova de s'Ònix | | |
| | Coves del Pirata | | |
| | Cova des Serral | | |
| | Coves del Drac | | |
| | Cova de sa Tortuga | | |
| Sardinia, Italy | Grotta di Nettuno | POS, SVS | Tuccimei et al., 2007 |
| Italy | Infreschi Cave | SVS | Bini et al., 2020 |
| Cuba | Santa Catalina Cave | POS | De Waele et al., 2017; 2018 |
| Bermuda | Government Quarry Cave | SVS | Harmon et al., 1978; 1981 |
| | Bierman Quarry Cave | | |
| | Crystal Cave | | |
| | Wilkinson Quarry Cave | | Wainer et al., 2017 |
| Yucatan Peninsula | Caves in Quintana Roo | SVS | Moseley et al., 2013 |
| Krk Island, Croatia | U Vode Pit | SVS | Surić et al., 2009 |
| Lošinj Island, Croatia | Medvjeđa spilja Cave | SVS | Surić and Juračić, 2010 |
| Andros Island, Bahamas | Blue Hole in South Bight | SVS | Gascoyne et al., 1979 |
| | Stargate Blue Hole | | Richards et al., 1994 |
| | | | Smart et al., 1998 |
| | Conch Sound Blue Holes | | Gascoyne, 1984 |
| Grand Bahama, Bahamas | Lucayan Caverns | SVS | Lundberg and Ford, 1994 |
| | | | Richards et al., 1994 |
| | Sagittarius Cave | SVS | Richards et al., 1994 |

**2.2 Spatial coverage.** Phreatic carbonate overgrowths on speleothems dating to MIS 5 are found in caves along the coasts of Mallorca (Vesica et al., 2000; Fornós et al., 2002; Dorale et al., 2010; Tuccimei et al., 2012; Polyak et al., 2018), northern coast of Cuba (De Waele et al., 2017; 2018), and to a lesser extent in Sardinia (Tuccimei et al., 2007). Submerged vadose speleothems were investigated from Bermuda (Harmon et al., 1978; 1981; Wainer et al 2017), Bahamas (Gayscone et al., 1979; Gascoyne, 1984; Li et al., 1989; Lundberg and Ford, 1994; Richards et al., 1994; Smart et al., 1998), Yucatan Peninsula (Moseley et al., 2013), Croatia (Surić et al., 2009; Surić and Juračić, 2010), and Italy (Tuccimei et al., 2007; Bini et al., 2020). Except for the SVS from Yucatan Peninsula reported by Moseley et al. (2013) and from Bermuda (Wainer et al., 2017), studies do not report the exact cave location from where samples were collected. Hence, the latitude and longitude for these indicators were determined using Google Earth to match locations from publication maps and noted accordingly. The geographical distribution shows all sites are located in the Northern Hemisphere within 20 and 45 ° latitude N (Fig. 2).

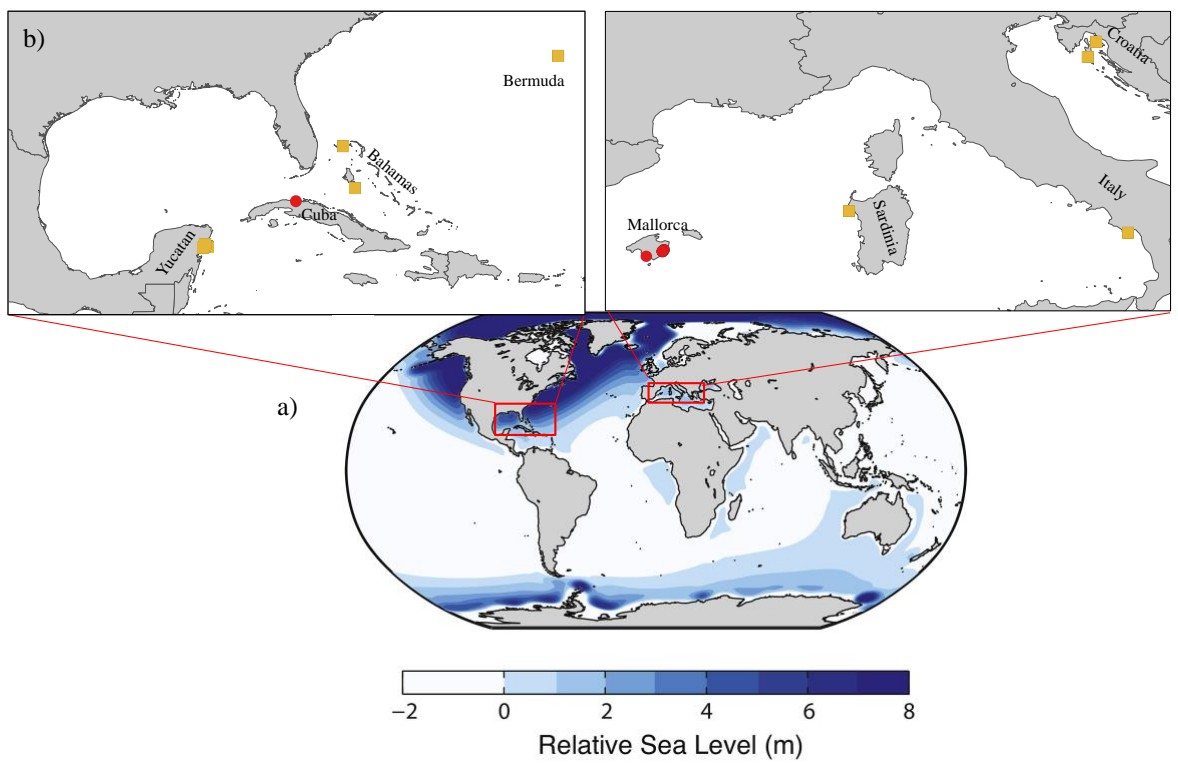

Figure 2. a) Global map showing the predicted RSL near the end (119 ka) of the modeled MIS 5e (see figure 2 of paper: https://doi.org/10.1016/j.quascirev.2017.06.013, Dendy et al. (2017)) to highlight the potential of the samples' sites to record the behaviour of the forebulge during ice-sheet loading and unloading. Sites where POS and SVS were documented and sampled are indicated by red circles and yellow squares, respectively.

**2.3 Elevation measurements and their uncertainties.** The sample elevation is the vertical distance between the indicator and a vertical datum (i.e., a 'zero' reference frame, representing modern mean sea level) and is a fundamental property measured in the field. All samples have been reported relative to present mean sea level. Several instruments with different uncertainties have been used to measure the elevation of the speleothems: barometric altimeter ($\pm$ 0.1 m; Moseley et al., 2013), metered tape or rod ($\pm$ 0.5 m; Harmon et al., 1978), and/or inclinometer ($\pm$ 0.05 m; Dorale et al., 2010). The uncertainties of Mallorcan POS elevation, even though some of the samples are collected from the same cave, vary depending on the instrument used (e.g., $\pm$ 0.1 m in Tuccimei et al. (2007) and Polyak et al. (2018); $\pm$ 0.05 m in Dorale et al. (2010)). The POS belt from Cuba occur within 0.4 m, more or less corresponding to the average tidal range measured in the city of Matanzas ($<$ 25 km NE from the cave site). In some cases, information regarding elevation measurements and their uncertainties are not reported because the papers are not strictly sea-level studies, but indirectly provide sea-level information, or because the uncertainties related to local tectonics are much larger than the measurement uncertainty (Surić et al., 2009). Samples reported by Gascoyne (1984) are missing information on their elevation, hence we only report their U-series ages in the database, and they are not considered

terrestrial limiting points. However, we include here the existing elevations of these samples found in the McMaster University Speleothem Collection, 2008 (courtesy: Derek C. Ford): samples 81048, 81049, 81050, 81052 from ~ 80 m below sea level, 81054 and 81055 from ~3 mapsl, 76017 from shallower depth, 78026 and 78027 from between 0-9 m below sea level.

**2.4 Samples.** In the WALIS database, each sample is assigned a *Sample ID* that is composed of the first two letters of the main author' last name, followed by the year of publication and a number code given to each speleothem individually (see Table 2 as an example). The *Analysis ID* complements the Sample ID by including a distinct number for when two or more ages are reported for the same speleothem. Finally, the *Reported ID* is the published sample identifier or Laboratory ID offered by the authors in the original paper. Since samples once collected may end up having several IDs (collection, dating lab, etc.), it is recommended that whenever included in a database, authors should always use the ID associated with its first description and when the Lab ID is different, this should also be added. We note that for samples that have different Laboratory ID than the Sample identifier (i.e., Moseley et al., 2013) both are included in our Reported ID.

**Table 2. Example to show how samples are coded in the WALIS database.**

| Sample ID | Analysis ID | Reported ID |
|---|---|---|
| DO10-001 | DO10-001-001 | CCVA-1 |
| DO10-002 | DO10-002-001 | CCVA-2 |
| DO10-003 | DO10-003-001 | CDD-1 |
| DO10-004 | DO10-004-001 | CDD-2 |
| DO10-005 | DO10-005-001 | CCVB |
| DO10-006 | DO10-006-001 | CCF-1 |
| DO10-007 | DO10-007-001 | CCF-2 |
| DO10-008 | DO10-008-001 | CPV-1 |
| DO10-009 | DO10-009-001 | CPV-2 |
| DO10-010 | DO10-010-001 | CPV-B8 |
| DO10-011 | DO10-011-001 | CPV-B6 |
| DO10-012 | DO10-012-001 | CPV-B9 |

A variety of different speleothems are reported to have recorded past sea levels. Stalagmites grow from the floor upward in caves, stalactites (dripstones) hang from the ceilings, and flowstones are sheetlike deposits that form on walls and floors. The POS analyzed by Dorale et al. (2010) and Polyak et al. (2018) have different size, shape, and morphology, depending on the vadose support and on the extent of their immersion in the brackish pool (Ginés et al., 2012). They are generally small enough to collect in-situ, and larger POS can be cored in place. The Cuban POS investigated by De Waele et al. (2017; 2018) in Santa Catalina Cave are rare composite mushroom-shaped speleothems with caps reaching diameters > 1 m and balconies present along the walls. In this case, the authors analyzed samples from mushroom stalk or cap, balcony, and other associated

speleothems. Most of the SVS investigated as sea-level indicators are stalagmites, but results from flowstones and stalactites have been reported as well.

The caves have unique scientific, recreational, and scenic value; hence, speleothems sampling strategies must be selective and reconcile cave conservation with the scientific goal (Baeza et al., 2018). For this reason, we emphasize that the utmost effort should be made to minimize the future impacts of sampling by following the conservation and preservation guidelines. We strongly encourage that great care must be paid to more sustainable sampling strategies and, if sufficient material and appropriate documentation has been archived by the original authors, making them available for future researchers is desirable. Similarly, we recommend to preferentially sample already broken/damaged speleothems whenever the original location can be established, as suggested by Frappier (2008). Another sustainable sampling strategy is coring of speleothems rather than collecting the entire specimen and patching the drill holes (Spötl & Mattey, 2012).

**2.5 Sample mineralogy.** The geochemical setting and sample mineralogy may dictate the susceptibility to alteration. For example, samples that show conversion of aragonite to calcite or calcite recrystallization, could have been subjected to uranium loss, which is an important factor that impacts U-Th ages (Lachniet et al., 2012; Bajo et al., 2016). To allow recognition of diagenetic fabrics, XRD screening is desirable for dating purposes (aragonite is preferred vs calcite). In order to make the best selection of samples, we encourage the use of petrographic investigation as well. Thin sections can help to best identify the speleothem layers just below and above the hiatus for dating purpose, and also, they reveal the internal structure and so, areas affected by recrystallization can be avoided while sampling. However, only some of the studies compiled here report the mineral assemblage of the samples by X-ray diffraction (Surić et al., 2009; Surić and Juračić, 2010; De Waele et al., 2017; 2018). De Waele et al. (2018), for example, complemented the screening method with petrographic investigations (thin sections) and imaging using scanning electron microscopy. We do not exclude the possibility that screening was performed in the other publications, but it has not been reported. For future studies, we strongly recommend including information on mineral assemblage, as well as diagenetic and crystalline descriptions.

**2.6 U-series methods**. Accurate geochronology is essential to these studies. Dutton et al. (2017) provide a template of what information should be reported for U-series data for geochronology and timescale assessment in the earth sciences. As was found for coral-based sea-level work, the speleothem U-series results reported often contain insufficient information to completely assess the data collected. This ultimately limits the value of the data since often it is not possible to assess or recalculate a date using the information provided. While it may not be practical to recalculate dates from older data, what is more important is to have data and methods so that decisions can be made to request or recollect samples of interest. We include in the database the U-series method used (alpha counting, thermal ionization mass-spectrometry (TIMS), multi-collector inductively coupled plasma mass spectrometry (MC-ICP MS)), the decay constants used, and the measured isotopic ratios and concentrations of the sample, all of which are required to recalculate U-Th ages and to assess their robustness. Generally, analyses of well-characterized internationally-recognized standards or certified reference materials are used to

demonstrate the reliability of results. Advances in technologies since the mid 1980s have notably increased the analytical precision and reduced age uncertainties, thereby allowing for dating of smaller sample-sizes, which permits better sampling along single growth layers (thicker samples will integrate material of different ages). The development of TIMS and then MC-ICP MS in measuring U-series isotopes constituted a major step forward from the alpha spectrometric method (Hoffmann et al., 2007). The majority of the records compiled here were dated using MC-ICP MS, and only few used or reported alpha spectrometry (Harmon et al., 1978; 1981; Gascoyne et al., 1979, Gascoyne, 1984; Smart et al., 1998; Vesica et al., 2000; Tuccimei et al., 2007) or TIMS results (Lundberg and Ford, 1994; Richards et al., 1994; Tuccimei et al., 2007; Dorale et al., 2010).

When provided by authors, the database also includes the initial $^{230}Th/^{232}Th$ ratios and the decay constants. The correction for the initial non-radiogenic sources (i.e., hydrogenous, colloidal and carbonate or other detrital components; Richards et al., 2012) of $^{230}Th$ incorporated at the time of speleothem deposition is extremely important for age calculation and is sensitive for samples that contain very little uranium or an abundance of detrital thorium. The $^{230}Th/^{232}Th$ activity ratio of 0.825 with an arbitrarily assigned uncertainty of 50 % found in the mean bulk Earth or upper continental crustal has been commonly assumed for initial $^{230}Th$ corrections. However, several studies have shown that this value may not cover all situations. Therefore, laboratories apply different corrections for the non-radiogenic detrital $^{230}Th$ fraction through either direct measurement of sediments associated with speleothems (Hoffmann et al., 2018) or computed isochron methods and stratigraphical constraints (Hellstrom, 2006; Richards et al., 2012). Most POS included in this database fulfill the criterion suggested by Hellstrom (2006) that samples with ratios of $^{230}Th/^{232}Th$ higher than 300 are considered clean, with very few samples having lower such ratio values (Tuccimei et al., 2007). However, the use of this threshold value is arbitrary and depends on the ratio used for initial Th. We emphasize that this database includes the ages as they are reported in the original publications and therefore, it contains two columns: reported ages and corrected ages. The latter one represents the ages to which the original authors applied a correction for detrital Th based on a priori estimate of initial Th. We did not apply any further corrections when compiling this database, and we also did not recalculate the ages. All data contains the minimum required information (as recommended in the supplemental material of Comas-Bru et al., 2020) to calculate uncorrected ages, however, not all of them provide the initial $^{230}Th/^{232}Th$ activity ratio to allow the calculation of detritus corrected ages. The decay constants for $^{234}U$ and $^{230}Th$ used to calculate the U-Th ages have been updated and improved over time (Jaffey et al., 1971; Edwards et al., 1987; Cheng et al., 2000; 2013). Most recent papers used the $^{234}U$ and $^{230}Th$ decay constants of Cheng et al. (2000; 2013), but in some papers, especially in the older ones, these values are not provided. It is not likely that published U-Th dates will be corrected and reported, but rather that assessment of materials and their U-Th dates might be used to make decisions on the need to obtain those samples through request or recollect samples from the same outcrop or site. Identifying and defining continuous growth of vadose speleothems could be helpful to refine minimum or maximum sea level elevations. However, continuous growth is strongly dependent on the reliability and resolution of the U-series chronology.

The datum for modern reference state of ages reported is either BP (or AD 1950) or it is not mentioned, in which case the ages are assumed to be calculated with respect to date of analysis. This is usually not considered significant because the errors on MIS 5e ages are generally > 500 years. However, with the improvements made using MC-ICP MS (Cheng et al., 2013), ages on high quality samples (i.e., aragonite mineralogy, high U content, insignificant $^{230}$Th correction) of last interglacial are now possible to uncertainties of ± 100 years (2σ), making how the age is reported more important (i.e., BP). Another reference

commonly used now is yr b2k (years before AD 2000). Except for Gascoyne (1984) and Gascoyne et al. (1979) where uncertainties are reported as 1σ, and Harmon et al. (1978 and 1981) who reported standard error, or errors were not mentioned, all ages are reported with 2σ absolute uncertainties. We refer the reader to the guide on the evaluation of the ages' quality which can be found in WALIS' official documentation at: https://walis-help.readthedocs.io/en/latest/RSL_data.html. We strongly recommend the use of the term "uncertainty" which is more appropriate in this context than "error", as suggested by

Dutton et al. (2017).

## 3 Discussions

The elevation of a sea-level indicator is not always coincident with the position of relative sea level at the time of its formation, but rather is correlated to it by a quantifiable relationship. This is defined by the indicative meaning, a concept that needs to be considered when calculating the elevation of paleo RSL (van de Plassche, 1986; Shennan, 2015; Hibbert et al., 2016; Rovere

et al., 2016a). To define the position of past sea level over space and time, the sea-level indicators need to provide information on the geographic positioning, elevation with respect to a contemporary tidal datum, age of formation, and the indicative meaning (Shennan, 2015; Khan et al., 2019). Most records of MIS 5 sea level have come from coral reefs, but the interpretation is hampered by the challenges of finding pristine and well-preserved corals and to the uncertainties related to the water depths above the corals. The past sea-level position in space and time indicated by speleothems depends on their type: i) POS have

the ability to define the discrete position (Fig. 3), whereas ii) SVS provide only an upper bound (Fig. 4).

### 3.1 POS define reliable positions of RSL

POS precipitation was tentatively associated with past sea-level stands almost five decades ago (Ginés and Ginés, 1974), but due to advancements in the U-Th dating, only recently have studies demonstrated their suitability as meaningful sea-level index points (Vesica et al., 2000; Dorale et al., 2010; Tuccimei et al., 2012; Polyak et al., 2018; Dumitru et al., 2019; 2021).

The most important benefit of using POS as sea-level indicators is their unambiguous relationship to sea level. For example, the indicative range accounts for the vertical extent over which an entire POS forms, and the reference water level corresponds to their thickest part, which is the midpoint of the indicative range and the mean sea level. Hence, they provide a unique opportunity to further enhance our knowledge on MIS 5 sea-level history, as presented below.

Obtaining 43 new U-series dates on POS from cave sites along the southern and eastern coasts of Mallorca, Polyak et al. (2018)

reported an accurate timing of MIS 5e sea-level history, with uncertainties of their U-Th ages better than ± 500 years. The

external elevation error within each cave is reported as ± 0.25 m and between all of the caves used in their study is reported to be ± 0.75 m. Their results show that relative sea level in Mallorca was ~2.15 ± 0.75 meters above present level (mapsl) between 126.6 ± 0.4 and 116 ± 0.8 ka, although centennial-scale excursions cannot be excluded due to some small gaps in the POS record (Polyak et al., 2018). Similar encrustations were found in Grotta di Nettuno, Capo Caccia area (NW Sardinia, Italy), ~500 km east of Mallorca, but only one episode of high sea stand at 4.3 mapsl was documented (Tuccimei et al., 2012). Polyak et al. (2018) attributed the discrepancy in RSL elevation between Sardinia and Mallorca to minor differences in glacial isostatic adjustment and/or tectonic movement at the two sites. Given the large age uncertainties obtained using alpha counting and TIMS (see Figure 3), it would be worth revisiting the chronology from Grotta di Nettuno and complement it with more samples. Two POS-derived MIS 5e data from Mallorca were also reported by Dorale et al. (2010) at 2.6 mapsl in Cova des Pas de Vallgornera at 116.2 and 120.6 ka. Using younger POS from the same and other caves, Dorale et al. (2010) showed that RSL in the western Mediterranean was at ~1 mapsl during MIS 5a (~81 ka), challenging the prevailing view of a much lower MIS 5a sea-level position. In summary, POS are hence reliable and accurate indicators able to better resolve RSL during MIS 5.

Over 200 large POS, out of which 24 were recently dated, occur in a limited altitudinal range of ~40 cm in Santa Catalina Cave on the northern Cuban coast (De Waele et al., 2017; 2018). The authors suggest that the oldest age (126 ka, sample SC2.6a) is the most likely to reflect the chronology of the MIS 5e sea-level stand, while the younger ages (which show much higher initial uranium activity ratios) reflect recrystallization processes. The present-day elevation of these 24 POS is at 16 mapsl (De Waele et al., 2017). These samples formed during a time interval when the sea level was the highest and the authors argue that slow uplift of the coastal area after their formation brought them to this elevation (De Waele et al., 2017; 2018). Additionally, these 24 POS contributed to the reconstruction of the speleogenetic stages and the local coastal uplift, while also providing information on the sea-level variations during the last ~400 ka.

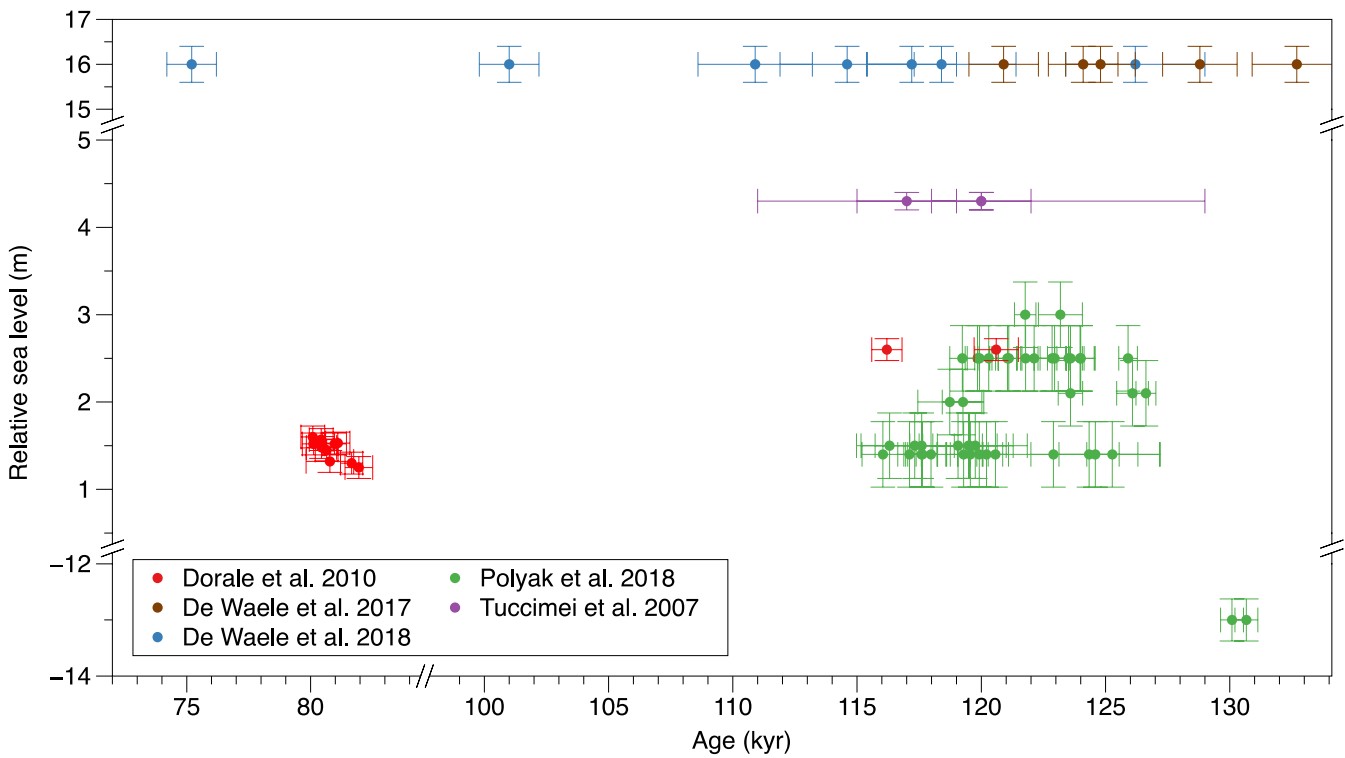

**Figure 3. Paleo RSL position recorded by POS. Except for Tuccimei et al. (2007), all U-Th ages are corrected for detrital Th as per the original publications. No other corrections such as tectonic movement or GIA have been applied.**

### 3.2 SVS growth intervals - implications for sea-level upper bound estimates

SVS from submerged caves in tectonically stable areas provide an additional source of sea-level data, as their growth stops when rising sea levels flood the caves (Richards et al., 1994). The timing of initiation and cessation of speleothem growth is fairly easy to resolve applying U-Th measurements on carbonate material extracted from above and below a growth hiatus. The obtained ages will indicate when the cave was air-filled and when the cave was submerged (Li et al., 1989; Richards et al., 1994). One advantage of using SVS is that they allow estimation of past low sea-level stands (lower than today) in both

interglacial and glacial periods. Such information is especially critical in tectonically stable regions (e.g., Bahamas and parts of the Mediterranean Basin), where changes in sea level can be directly associated with changes in ice-sheet volume, which is a challenging task for sea-level indicators found on the uplifting coastline regions (e.g., Huon, Barbados, Haiti).

Submerged speleothems from Bermuda provide strong sea-level evidence and contribute notably to deciphering the timing of stillstands during MIS 5. The growth deposition of stalactites, stalagmites, and flowstones with low $^{232}$Th content found in

Bermudan caves indicates that sea level stood at ~4 to 6 mapsl at ~125 ka, it fell below −6.5 m by 120 ka, and it stabilized for

a short time at approximately 8 m below present at ~114 ka (Harmon et al., 1978). A few years later, another study by Harmon et al. (1981) using ages from submerged speleothems and eolianites showed that sea level at ~105 and ~85 ka was between −15 and −20 m and at least −15 m at ~95 ka. A recent research on a submerged stalactite from Wilkinson Quarry Cave, northern Bermuda, indicates hiatuses starting at 137 ± 5, 106 ± 6, 84.3 ± 1.5, and 79.9 ± 0.9 ka (Wainer et al., 2017). These new results provide important constraints suggesting that local sea level might have peaked above ~1.5 mapsl during MIS 5e, 5c, and 5a. They also indicate a double highstand during MIS 5a, and while the data are in agreement with other studies from tectonically active regions (Surić et al., 2009), given the tectonic stability of Bermuda, Wainer et al. (2017) interpret the results as a global event associated with changes in ice-sheet volume, and not a tectonic movement.

A large number of speleothems were collected from the Bahamian caves from different elevations as mentioned earlier in Section 2.1, but only a few of them captured sea-level stands during MIS 5. Spalding and Mathews (1972) first attempted to date SVS from Grand Bahama Island and were followed by Gascoyne et al. (1979), Gascoyne (1984; northeastern coast of Andros and Lucaya Caverns, Grand Bahama), Richards et al. (1994; South Andros and Grand Bahama) and Smart et al. (1998; South Andros). Li et al. (1989) first presented ages on a flowstone collected from −15 m below modern sea level in Lucayan Caverns, Grand Bahama Island and documented hiatuses at a few different intervals including 133−110 ka and 100−97 ka. These ages were substantially improved by an order of magnitude using TIMS method and thus, sea level during MIS 5a was constrained to be below −8.5 m (Lundberg and Ford, 1994) and between −15 m and −18 m, as suggested by Richards et al. (1994). The latter study also precisely constrained the termination of MIS 5a by determining the age of a sample that started growth at 79.7 ± 1.8 ka. More recently, Richards et al. (2012) revisited these estimates and applied the correction for non-bulk Earth initial $^{230}Th/^{232}Th$ contamination using the isochron method on samples of stalagmites from the same settings (Richards and Dorale, 2003). Stalagmites from a blue hole east of Andros Island were also sampled and dated for sea-level application. The first set of ages showed that some of these samples deposited during MIS 5; however, they were found to be out of stratigraphic order possibly due to incorporation of minor amounts of marine deposits surrounding the central core (Gascoyne et al., 1979). The authors eliminated the age-biasing marine deposits by progressive acid leaching of crushed samples and analyzed the cleaned calcite crystals and the new results showed higher ages documenting a MIS 6 sea-level lowstand (at least 42 m below present level) between 160 and 139 ka (Gascoyne et al., 1979).

Subaerial growth periods of speleothems from Yucatan Peninsula provide precise and reliable maximum constraints for relative sea levels during MIS 5 that contribute to our understanding of sea level in the western North Atlantic–Caribbean region (Moseley et al., 2013). The authors show that following the MIS 5e highstand, RSL dropped below −4.9 m by 117.7 ± 1.4 ka. They also provide maximum sea-level stands during MIS 5c and 5a: i) MIS 5c sea-level highstand occurred after 107.7 ± 0.9 ka between −11 and −4.9 m and ended by 108.2 ± 4.9 ka and ii) sea level peaked during MIS 5a after 87.6 ± 0.6 ka at an elevation higher than −9.9 m, but the length of this highstand is poorly constrained because of the high detrital Th in the speleothem. Sea level then dropped below −14.6 m and the speleothem continued to grow continued until 61.3 ± 0.4 ka. In addition to the information provided on the maximum RSL, when speleothems with precise ages are used in conjunction with

data from other local sea-level indicators, they can also offer robust chronologies for the timing of the relative sea-level fall at
the last interglacial termination (Moseley et al., 2013).

Constraints on the maximum RSL during MIS 5a in the eastern Mediterranean basin are provided by Surić et al. (2009) who
dated two submerged stalagmites from the U Vode Pit on the Krk Island, in the eastern Adriatic Sea. Their double highstand
MIS 5a sea-level scenario is supported by layers of halite and gypsum associated with hiatuses in speleothem growth probably
caused by seawater inundation. The timing of possible marine incursions was determined by dating the layers below and above
these growth hiatuses. Their work was further continued by Surić and Juračić (2010) who investigated 16 submerged
speleothems from 7 submarine caves and pits along the Eastern Adriatic coast to provide insight into the sea-level changes
over the last 220 ka.

Reconstructions of MIS 5e sea-level evolution are also provided by stratigraphical and geochoronological constraints from the
Infreschi Cave (Marina di Camerota, Italy). A combination of U-Th ages of speleothem deposition phases with ages of calcite
in Lithophaga boreholes and with $^{40}$Ar/$^{39}$Ar ages of correlated thephra indicates that sea level fell more than 6 m before ~120
ka, and places the maximum highstand RSL at 8.90 ± 0.6 mapsl (Bini et al., 2020).

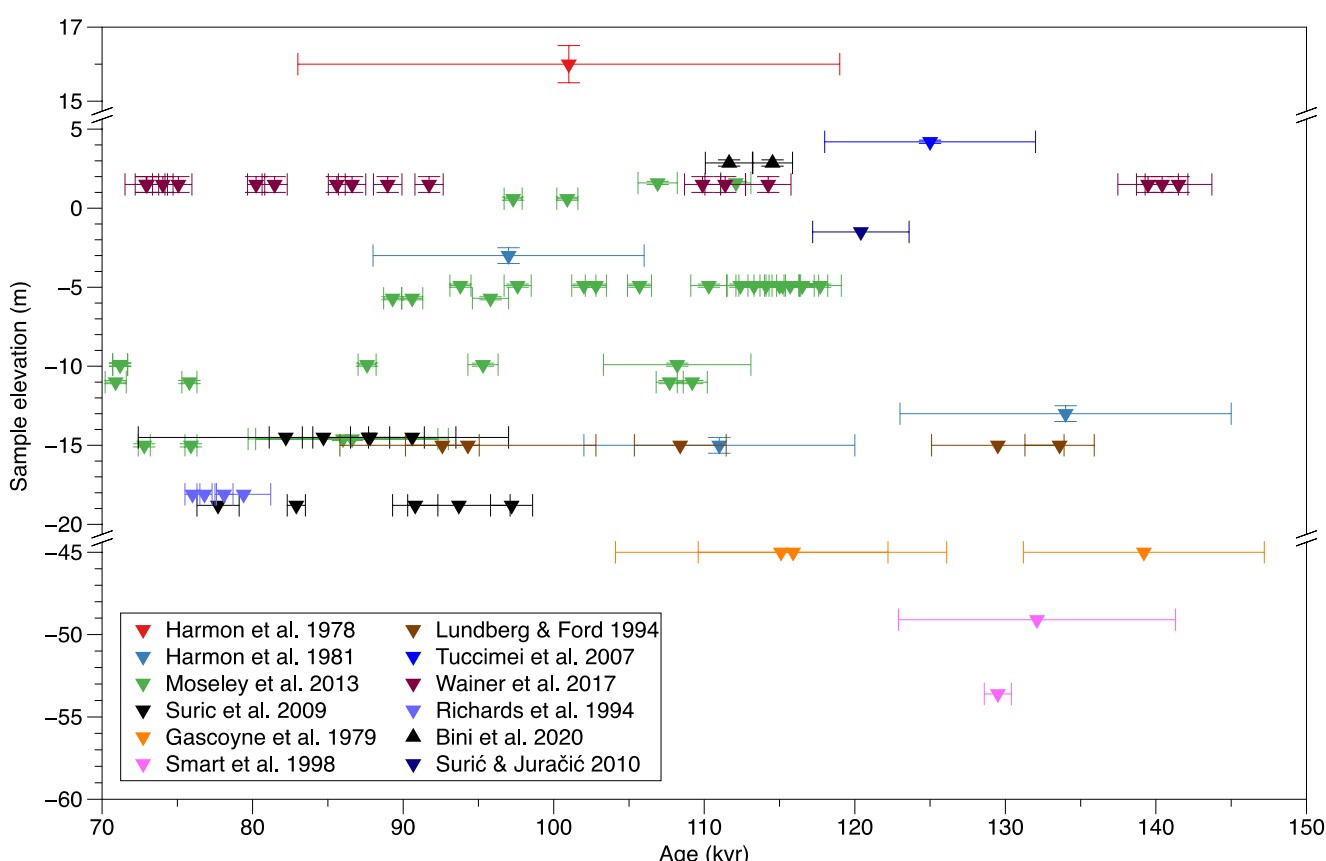

 **Figure 4. SVS elevations and U-series ages indicating times when RSL must have been lower, hence providing maximum RSL constraints. Note that continuous growth in some of these speleothems could provide more representative results. None of the data are corrected for GIA or long-term deformation.**

### 3.3 POS- and SVS-derived sea level provides inputs for glacial isostatic adjustment models

The recorded elevations of sea-level indicators reflect a combination of local changes in sea level of a region due to glacial isostatic adjustment that causes land uplift or subsidence, displacing the local sea-level datum relative to the global mean (Rovere et al., 2016b). Thus, a significant part of the spatial variation in the MIS 5 sea-level record is due to the diverse response of coastlines to GIA, depending on the extent of continental or insular shelves, the distance from former ice sheets, and other variables (Dutton and Lambeck, 2012; Creveling et al., 2015; Dendy et al., 2017). Consequently, interpreting local sea-level markers first involves understanding and correcting for the ongoing effects of GIA. Numerical models generally estimate GIA with assumptions about the thickness, distribution, extension, and duration of former ice loads as well as the viscoelastic properties of Earth (Mitrovica and Milne, 2003; Austermann et al., 2017). The speleothems from locations such as Mallorca, Bermuda, Bahamas, and Yucatan Peninsula provide an additional and independent source of sea-level data, which have the potential to provide valuable information about the deformation parameters of the Earth and about the distribution of past ice sheets. Their distance from large pre-existing ice sheets and relatively stable passive margin tectonic setting is what makes them ideal locations.

The high-resolution POS-derived sea-level record from Mallorca was corrected to estimate the ice-equivalent sea level by using nine different GIA models (Polyak et al., 2018). The authors' preferred model showed that in Mallorca sea level peaked early in the MIS 5e at 5 mapsl due to the proximity to the forebulge of the penultimate glacial maximum (PGM) Eurasian Ice Sheet and then gradually decreased and stabilized by 122 ka at $2.15 \pm 0.75$ m, until the highstand termination at 116 ka. None of their models support the hypothesis of a second highstand during MIS 5e, and their results show no evidence for rapid sea-level fluctuations larger than 1 m. They also suggested that the corrected sea level is more sensitive to the size and distribution of the Northern Hemisphere PGM ice sheets than to the one-dimensional Earth model used for other predictions. This record allowed the authors to test both the sensitivity of MIS 5e sea-level stand predicted by GIA models to PGM ice distribution and the timing of deglaciation (Polyak et al., 2018).

Bermuda is strongly affected by the glacial forebulge that forms due to the presence of the Laurentide Ice Sheet during glacial periods (Harmon et al., 1981; 1978; Wainer et al., 2017), hence sea-level markers from this location have the potential to capture the response of the forebulge to glacial loading and can be seen as gauges. Based on U-Th ages of SVS from Wilkinson Quarry Cave, Bermuda, Wainer et al. (2017) showed that RSL was higher than $1.5 \pm 0.5$ m during MIS 5e, MIS 5c, and MIS 5a. The results also suggest a double RSL peak during MIS 5a, indicating rapid sea-level variation associated with changes in ice-sheet volume. Testing GIA models against these new constraints, the results reinforce the presence of a smaller Laurentide Ice Sheet during late MIS 5 and a restricted range of Earth viscosity values, with higher values of lower-mantle viscosity

compared to those used in some GIA models (e.g., Creveling et al., 2017). Wainer et al. (2017) confirm that it is possible to explain a wide range of MIS 5c–a relative sea levels observed across the Western North Atlantic–Caribbean in GIA models, even with a limited range of mantle deformation constants.

The Bahamas Archipelago spans the south-eastern edge of the peripheral bulge of the ancient Laurentide ice complex, and hence this is a region that captures gradients in the GIA prediction. The majority of reconstructed MIS 5 sea-level estimates in the Bahamas use corals (Chen et al., 1991; Thompson et al., 2011; Skrivanek et al., 2018; Muhs et al., 2020). While these records demonstrate the potential of this area for sea-level reconstruction, despite the data abundance, the precision of the inferred water depth and also the open-system U-series behavior of some corals, cast uncertainty on existing RSL estimates. Undoubtedly, the uncertainties behind GIA corrections and the models that investigators adopted, also lowers the accuracy of the reconstructed sea level. In addition to the GIA corrections, the long-term subsidence of the platform needs to be also considered when assessing the LIG sea-level record. Subsurface drill core data from shallow water stacked facies indicate that the Bahamas is subsiding at rates of several meters per 100 ky (McNeill, 2005). None of the Bahamian cave deposits reported so far (Gascoyne et al., 1979; Gascoyne, 1984; Lundberg and Ford, 1994; Richards et al., 1994; Smart et al., 1998) address these corrections, but observations indicate the abundance of this type of sea-level indicator and its potential to provide additional data to help decipher the duration and amplitude of LIG sea level in this region.

Moseley et al. (2013) used SVS growth periods to constrain maximum elevations of relative sea level, which are in agreement with GIA models for the near to intermediate region of the former North American Ice Sheet. Based on GIA modeling, Lambeck et al. (2012) classified Yucatan Peninsula as a near-field site in the western North Atlantic–Caribbean region located on the deformational bulge; thus, its MIS 5 sea-level indicators are affected by significant GIA effects. Moseley et al. (2013) provide additional constraints on the timing of sea-level fall following the MIS 5e highstand and on sea-levels peaks during MIS 5c and 5a, indicating that a second MIS 5a highstand did not reach as high as the first in the Yucatan. The authors highlight the challenges of comparing sea-level records from the Yucatan Peninsula with data even from other field sites in the western North Atlantic–Caribbean region due to the complexity of estimating the GIA effects. They also emphasize the need for improved estimates of the MIS 5 sea-level highstand in the Yucatan in order to better constrain the sea-level history that can be used in predictive GIA modeling studies.

Altogether, both POS and SVS provide powerful constraints for future GIA models and also help refining ice-sheet histories and solid Earth properties. The geological data can test the GIA models, which will lead to an improvement of the modeled MIS 5 sea-level elevations for a large number of localities around the world.

### 3.4 POS- and SVS-derived sea level register tectonic uplift rates

Long-term deformation (uplift or subsidence) can substantially affect the speleothems-derived sea-level results. Assessing POS ages and their elevations and knowing their clear relation to sea level, proved to be a useful tool in estimating past tectonic evolution of the regional coastal karst landscape (Fornós et al., 2002; De Waele et al., 2018; Dumitru et al., 2019).

Using U-Th data from Vesica et al. (2000), Fornós et al. (2002) suggested some tectonic tilting (increasing elevations northwards) in the eastern part of Mallorca based on POS horizons dated at MIS 5e, 5c and 5a. They estimated an average minimum velocity of the tilting of ~0.02 mm/year in the southern part with respect to the north. However, more recent higher-resolution U-Th results with better analytical precision and also more accurate elevation measurements provide evidence for tectonic stability of the region from MIS 9 to the present (Dorale et al., 2010; Polyak et al., 2018). This evidence is based on the presence of multiple highstands within the same cave indicated by different POS horizons, which suggest that RSL remained within the vertical extent of the cave over time (1.1 mapsl during MIS 9; 2.15 mapsl during MIS 5e, and 1.42 maspl during MIS 5a; Dorale et al., 2010; Polyak et al., 2018). A more recent study has confirmed the tectonic stability of Mallorca based on the observed elevation of six Pliocene POS and estimated a median uplift rate at this site of 0.002 mm/year (0.0006–0.0044 m/Myr; Dumitru et al., 2019).

POS sampled from Cuba reveal Upper Pleistocene–Holocene coastal uplift rates at Matanzas between 0.05 and 0.10 mm/year (De Waele et al., 2017), one order of magnitude lower than those reported previously based on geodetic measurements along various parts of the Cuban coastline (Iturralde-Vinent, 2003). These results confirm the lower uplift rates of the western part of Cuba compared with the eastern sectors and argue that uplift rates are site-specific (De Waele et al., 2017). De Waele et al. (2018) further show that coastal uplift of Cuba has varied widely over the last 600–400 ka, with no uplift or even periods of slow subsidence that characterized the MIS 11–MIS 5e time frame. Data also allowed to place these stages in a broader regional context of uplift and sea-level variations during the last ~400 ka.

The SVS collected from Croatia indicate MIS 5a sea-level stand of at least −14 m (Surić et al., 2009), however, the authors acknowledge that these results would most likely need to be corrected for long-term regional tectonic uplift of 0.15–0.25 mm/year with episodical subsidence events generated by collision of Adria microplate with Eurasia Plate (Surić et al., 2009). If other model evidence is assumed, these results can put constraints on temporal uplift.

## 4 Concluding remarks and future directions

This dataset paper represents the first compilation of speleothems (POS as sea level index points and SVS as terrestrial limiting points) derived sea-level history for the last interglacial period. The purpose of this work is to contextualize the interpretation of speleothems records in a framework that would facilitate the MIS 5 sea-level research community to use the worldwide database. Littoral caves offer a means of addressing the temporal and spatial sea-level data gaps in other proxies, by hosting deposits which provide an opportunity to independently date records of past sea-level changes. The phreatic overgrowth mechanism that deposits calcite/aragonite at sea level arguably provides the most precise and less ambiguous indicator of RSL timing and elevation. Intervals of SVS growth is indicative of times when sea level must have been lower than their elevation.

Hiatuses can also be used to indicate sea level position, however, they must be cautiously interpreted since there are several reasons that cause their occurrence. Hiatuses associated with biogenic encrustations or borings could potentially indicate more exactly where sea level was with respect to the SVS. Speleothems from locations such as Mallorca, Bermuda, Bahamas, and Yucatan Peninsula are particularly useful, not only for the information they provide about the eustatic sea level but more importantly for the critical constraints on future GIA models to help refine ice-sheet loads and regional mantle rheology. These records of sea level from low-lying islands and continental coastlines could benefit research related to other disciplines, such as water resources availability, sea-level rise projections, and saltwater intrusions.

Our contribution to the WALIS database should be seen as a highlighting point on the relevance of speleothems for sea-level studies. In this compiled database on MIS 5 sea level, we list the areas where these sea-level indicators are located and identify future research priorities. One research direction could be exploring for additional POS levels from Sardinia, which in conjunction with the extensive POS data from Mallorca would provide relevant information for better GIA and tectonic context for the western Mediterranean basin. Another direction could be sampling and dating the stalagmites identified at elevations much lower than present sea-level in caves from the Bahamas and Bermuda; their chronologies can contribute to an improved record of low sea-level stands during the Pleistocene. Of equal priority is to re-analyze the samples that have been dated by means of alpha spectrometry, using the more advanced facilities in order to increase the precision of the ages. Of great importance is exploring new locations for additional POS or SVS along continental and island carbonate coastlines that would complement the already existing records.

To build a more valuable dataset that will be used across different disciplines, we strongly encourage researchers publishing new sea-level studies based on speleothems to include the following information:

- Sea-level indicator and its relationship to sea level: i) site location (latitude and longitude of the cave); ii) the elevation of the sea-level indicator, the instrument type used and its precision, and the error associated with the elevation measurement (when using barometric altimeter or diver depth gauge for submerged samples, the elevations should be adjusted for density variation in the water column whenever salinity profiles are available, and the vertical density profile associated with fresh-water, brackish, and saline zones should be included with the depth information); iii) the sea level datum to which the elevations are referred and how the indicative meaning has been quantified.

- Screening results: XRD, petrography, and polarizing/scanning electron imaging, including information on mineral assemblage, as well as diagenetic and crystallization descriptions (e.g., fabric).

- U-series data: In order to collectively improve the utility of U-series data we encourage researchers publishing new speleothem-based sea-level studies to follow the recommendations suggested by Dutton et al. (2017) in reporting their data. These authors specify the required data to enable calculation and, if needed, re-calculation of the same ages using different parameters and also, to facilitate the interpretation in the context of other studies. The checklist of

minimum data to report includes: uncertainties for all parameters, state whether uncertainties on ages include decay constant uncertainties; Names, descriptions, and reference values of reference materials; Decay constants; Isotopes in spike and method of spike calibration; Method of calibration for all activity or atom ratios reported; Activity or atom ratios for $^{230}Th/^{238}U$ (or $^{230}Th/^{234}U$) and $^{234}U/^{238}U$; $^{230}Th/^{232}Th$ activity or atom ratio; Details of procedures and values used to interpret ages using isochrons or other models; Date of analysis or reference age (e.g., BP, b2k, etc.). These recommendations will increase the usefulness of this type of analytical results in the U-series geochronology community (Dutton et al., 2017). We also recommend reporting growth rate which allows to better define the onset and cessation of deposition for either POS or SVS samples.

## 5 Data availability

The speleothems database is available as open access and periodically updated as needed, at the following link: http://doi.org/10.5281/zenodo.4313860 (Dumitru et al., 2020). The description of each field in the database can be found at: https://doi.org/10.5281/zenodo.3961544 (Rovere et al., 2020). More information on the World Atlas of Last Interglacial Shorelines can be found at: https://warmcoasts.eu/world-atlas.html. Users of our database are encouraged to cite the original sources alongside with our database and this article.

## Author contributions

O.A.D. compiled the data, drafted, and wrote the manuscript with input from B.P.O., V.P., and Y.A. BPO contributed to designing the figures and V.P and Y.A. provided expert review of U-series data.

## Competing interests

The authors declare that they have no conflict of interest.

## Acknowledgements

The data used in this study were compiled in WALIS, a sea-level database interface developed by the ERC Starting Grant "WARMCOASTS" (ERC-StG-802414), in collaboration with PALSEA (PAGES / INQUA) working group. The database structure was designed by A. Rovere, D. Ryan, T. Lorscheid, A. Dutton, P. Chutcharavan, D. Brill, N. Jankowski, D. Mueller, M. Bartz, E. Gowan, and K. Cohen. The data points used in this study were contributed to WALIS by: Oana-Alexandra Dumitru, WALIS Admin, Alessio Rovere, Ann-Kathrin Petersen, Deirdre Ryan (in order of numbers of records inserted). We thank Vanessa Johnston, David Richards, and an anonymous reviewer for their careful feedback and useful recommendations.

Special thanks to Alessio Rovere for his guidance and assistance in the data compilation and manuscript preparation. We also thank Dr. Joyce Lundberg for providing the elevation information on the samples from McMaster collection. Part of the data included in the database results was generated by a collaborative NSF grant (AGS 1602670 and 1602685) to B.P.O. and V.J.P.

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
