# Peer review of "Last Interglacial (*sensu lato*, ~130 to 75 ka) sea level history from cave deposits: a global standardized database"

_Earth System Science Data, 2020_

## Referee Comment (RC1) · Vanessa Johnston (Referee) · 9 Jan 2021

General comments:

The dataset compiles chronologically constrained, paleo relative sea-level based on previously published proxy records from cave deposits. The dataset has been compiled using a standardized format as part of a wider project to collate worldwide paleo sea-level data. In this way, the data are presented in a format that is easily usable by researchers who are not necessarily experts in cave deposits. This unique dataset, focusing on cave deposits, is an important addition to the database of more commonly used sea-level proxies, such as corals. The data, extracted from the original publications, appear to be inserted in a thorough and complete manner.

The manuscript accompanying the dataset is informative and well written. It states clearly how these deposits form and their significance to reconstructing paleo sea-level, with the use of simple diagrams, enabling researchers from other fields to understand the formation processes of these unique cave deposits. Moreover, it discusses the limitations of these deposits for assessing the paleo sea-level, both temporally and spatially, and outlines the uncertainties on these values with respect to the data in the dataset. The discussion presents the data in simple graphical forms and remarks on the significance of these data for paleo relative sea-level, glacial isostatic adjustment, and tectonic uplift. As would be expected from a manuscript accompanying a dataset, the authors have made no attempt to analyze the data in terms of paleoclimate/tectonics but have outlined some of the conclusions from the original publications and identified research directions to improve and expand the dataset.

Specific comments:

The POS are said to form in the brackish waters of the seawater–meteoric water mixing zone (line 61). In general, thermodynamic models of the mixing between seawater and calcite-saturated meteoric water produce aqueous solutions that are under-saturated with respect to calcite, and thus, promote calcite dissolution. The authors later note that the POS only form with the fluctuating water table, which is a key concept from a geochemical perspective and could be easily overlooked. I feel that this explanation of POS formation should be slightly expanded to overcome problems that may arise when a reader might expect calcite under-saturation in this zone, rather than the carbonate precipitation that is evident from the formation of these cave deposits.

Figure 1: This figure provides a good overview of the formation of POS and SVS deposits. However, I find that part b should have a better explanation to explain more clearly what the arrows are showing. In part c, the graph is very small and difficult to read in comparison with the other parts of the figure. Furthermore, it is a conceptual

model but there are values on the y-axis; are these values referring to a specific case? Perhaps the x-axis also requires an arrow indicating "time". As a suggestion, the size of the orange oval (POS deposit) could be increased over time through points T1–T3 to show that it is growing. The inset photo in part c should probably include a scale and location information. Part d requires a scale.

Technical comments:

Ln 13 and throughout: Please check consistency of text between "sea level" and "sea-level".

Ln 32: Remove the word "former" as it is not needed because you have "paleo" in the sentence. Also, sea-level should perhaps be pleural since the sentence ends with "them".

Ln 36: Please add the word "period" after "the last interglacial".

Ln 40: Consider adding "partial", or similar, before the word "analog", as it is not a strict analog due to a different orbital configuration.

Ln 49: Please add the word "level" after the word "sea". Consider converting "still stand" to "stillstand" and, throughout the text, also consider the usage of "high stand or highstand", and be consistent.

Ln 52: The letter "L" in the word "AnaLysis" should probably be capitalized following the SISAL acronym.

Ln 61: Consider changing "sea water" to "seawater" and check consistency throughout the manuscript.

Ln 119: Please add the word "period" after "the last interglacial".

Ln 133: Perhaps quote "xls" as "Microsoft Excel .xls" (or indeed .xlsx), or similar.

Ln 135: Please change the word "which" to "that" (because "which" should almost

always be preceded by a comma).

Table 1: This table is a bit difficult to follow since the locations/deposit types have been placed central to the row to which they refer. Please format in a different way.

Fig. 2 caption: Please change "modelled" to "modeled" as you are using American English.

Ln 207: Please remove the word "times".

Ln 262: Please consider changing the word "high" to "large" [uncertainties].

Ln 265: It is unclear to what the word "respectively" is referring to.

Fig. 3 caption: Remove the full stop after the word "Figure".

Ln 284: Please insert parentheses around the "e.g.," clause. For example (e.g., your text here).

Ln 294: The dates quoted here require a citation in this sentence.

Ln 316: Please change the hyphen to an en-dash between "Atlantic" and "Caribbean".

Ln 318: Please remove the space between "5" and "c".

Ln 319: Perhaps it would be clearer to write "reaching above $-11$ m, but not as high as $-4.9$ m" as "between $-11$ m and $-4.9$ m".

Fig. 4: Change Harmon citation date from 1878 to 1978!

Ln 347: The word "authors" should be authors' (add apostrophe) because it is a possessive meaning being used.

Ln 348: Please remove the words "the former".

Ln 363: Please use an en-dash between the "5c" and "a" (i.e., 5c–a).

Ln 385: Please change "refining" to "to refine".

Ln 401: It is unclear to which island you are referring (Mallorca or Sardinia).

Ln 402: Please remove spaces either side of the en-dash.

Ln 403: Please use an en-dash, rather than a hyphen, between the words "Pleistocene" and "Holocene".

Ln 407: Please use an en-dash between the values 600 and 400 ka.

Ln 408: Please change hyphen for an en-dash and remove spaces before and after to become "MIS 11–MIS 5e".

Ln 415: Please add the word "period" after "the last interglacial".

Ln 424: Please consider removing the "etc." and changing the sentence to a phrase, such as "... benefit research related to other disciplines, such as, water resources availability, sea-level rise, and saltwater intrusions."

Ln 430: This line states "elevations much lower than present in caves", I presume you are referring to present sea-level, please then add the word "sea-level" into this phrase.

---

## Referee Comment (RC2) · Anonymous Referee #2 · 15 Jan 2021

I have now completed my review of the manuscript of Dimitru et al. I think it is an useful contribution with several interesting considerations, which would be further expanded in the future works. Considering I'm not an expert of the WALIS database I don't give feedback on the database itself even if the most relevant information seem to have been considered. I have several points along the text which I can summarise in some general comments:

- From the text there is no a clear discriminant petrographic, geochemical and morphological criteria to separate POS from other speleothems which forms in freshwater pools in caves. I think this would be very useful to give this description to clarify

that these are unique speleothems typology. - SVS are basically terrestrial limiting points. The use of hiatus as indicator of sea level submersion in absence of other clear evidences is probably dangerous and risk to be overexploited (hiatuses form in spelethems form many reasons). This is clearly stated in the manuscript. But then is not considered anymore as criteria and probably a more critical approach for previous data should be considered. - Detrital correction in speleothem dating is critical. Probably a short mention on the fact that different laboratories applies different correction is useful (also in the past). It would be useful to have a range of 232Th/230Th values found is POS speleothems for the reader in the text. - Along the text there are many sentences unclear and/or not very precise (in my opinion). I have suggested modification. - Some sentences need to be supported by references. Overall the manuscript is easy to read (even if not all sentences are consequential) and general conclusions interesting even if probably (considering the nature of the special issue and that of the manuscript) a section including some general more methodological approach in selecting material and advices as use data for some conclusion would be useful.

Specific comments (generally minor)

Line 40 "….(Capron et al., 2019), and MIS 5e is considered an analog for the Holocene." The Holocene is quite different in terms of insolation and sea-level history. Please avoid oversimplification. MIS5e is not a good analogue for the Holocene, but it is the closest interglacial we can study with relatively good details. The main point is that MIS5e has temperature higher than pre-industrial Holocene, but other boundary conditions are very different. Line 43 "…uncertainties in the reconstructed sea level." Please Insert a reference

Lines 42-43 "Fossil corals can be dated to relatively high precision but have meter scale uncertainties in the reconstructed sea level" Please insert references

Lines 43-44 "Other indicators such as erosional notches pinpoint sea level, but lack age control". Please adds citation Are you sure you want refer to erosional notches (they

are not the best, but in case I can suggest Bini et al. 2014 Earth Surface Processes and Landforms, 39 (11), 1550-1558). Note that this is the same problem for tidal notches, probably some like erosional sea level markers are difficult to be dated? Or? Some more general?

Lines 44-45 For this reason, there is a growing demand in exploring additional sea level indicators that can complement the information derived from fossil corals, while simultaneously having robust chronology.

Maybe the sentence is not completely true most of the indicators are well known but the point is to clarify the indicative meaning and to date correctly them.

Line 46 "...as coral reefs (Thompson et al., 2011)..." are these unique of karst environment?

Line 50 "....submerged vadose speleothems (SVS; suggesting maximum elevations of sea level position)."

Not very clear. Why they should be maximum elevation? They indicate the cave inundation (see your line 80) or air filling, so they are terrestrial limiting points. I think we must be conservative in these concept or be more clear in the explanation I have the same observation to other points

Line 95 " was actually located throughout the bulk of the rise-fall cycle (Richards et al., 1994; Surić et al., 2009). Therefore, it can be difficult interpreting the relationship between vadose speleothems growth and sea-level history."

Yes, I agree. Translating in a few word they are terrestrial limiting points? Is that you want to say? Probably this concept can be introduced here considering the general nature of the special issue, to avoid to return to the same concept later. I think this point should highlighter later in the conclusion or in a special section where it should be stated that use of SVS hiatus need to be considered with cautions. Maybe showing some examples.

Lines 166-170 "Several tools with different uncertainties have been used to measure the elevation of the cave deposits: barometric altimeter ($\pm$ 0.1 m; Moseley et al., 2013), metered tape or rod ($\pm$ 0.5 m; Harmon et al., 1978), inclinometer ($\pm$ 170 0.05 m; Dorale et al., 2010)." I think here there is some confusion between instrumental precision and accuracy of the measure. Not clear. For instance barometric altimeter can give you this kind of precision, but it is quite far from the accuracy of the elevation measurement. All of this have been reported to local datum? Just to clarify.

Line 175 "….or that precise measurements are not so relevant because the uncertainties 175 related to local tectonics are much larger (Surić et al., 2009)." Sentence unclear precise measurements are mandatory. Maybe delete?

Section 2.5 "Given that speleothems are less likely subjected to alteration and diagenesis when compared with organic precipitation of corals, an elaborate sample prescreening is not critical. Thus, only some of the studies compiled here report the mineral assemblage of the samples by X-ray diffraction (Surić et al., 2009; De Waele et al., 2018). However, we cannot exclude the possibility that screening was performed, but not reported."

This is misleading section. XRD are fundamental in corals because they are aragonitic and tend to transform in calcite. Most speleothems are not aragonitic but calcitic. However, you stated at line 420 (why not before) that "The phreatic overgrowth mechanism that deposits calcite/aragonite at sea level…." So the problem for POS is real, and aragonite risk to be recrystalised all the same. However, it is a fundamental practice in speleothem science to select samples after a screening in thin section because also calcite can experience open-system behaviour. So this sentence should clearly state that (especially for POV and SVS) a check for potential alteration is necessary. A good case is the discussion made by De Waale et al 2017,2018, but for vadose speleothems there are many useful references for checking potential alteration. I think this section should be rewritten. I think what is lacking here which are the criteria to cite that POV are really POV and not formed in a lacustrine environment. I think there are not a very

precise description here to discriminate them and to charaterised better POV.

Line 201-202 "While not always necessary, including information on mineral assemblage, and diagenetic and crystalline descriptions are useful."

I think this is a mistake and I don't' think is the though of the writers. Petrographic and diagenetic description is fundamental and mandatory!

There is no mention here on the clastic contamination (but it is later). Presumably in POV would be minor but according the compilation proposed it would be nice to have a range here of measured 232Th/230Th and eventually to give a range of correction performed by different lavoratories. This can have an effect on the final age.

Line 215 "The development of TIMS and then MC-ICP MS in measuring U-series isotopes constituted a major step forward from the alpha spectrometric method." Maybe here a reference is necessary.

Line 223-224 "Hellstrom (2006) suggested that a ratio of 230Th/232Th larger than 300 can be considered as an indicator of clean samples not requiring correction for detrital thorium." It is probably useful to mention that different laboratories applies different correction (e.g. Bulk upper crust or iterative calculation of the initial ratio) (see also previous point).

Line 232 "…..age errors are now possible to $\pm$ 100 years…," yes, but this kind of error for speleothems is probably just an analytical error. Considering clastic contamination and growth rate to have such a high "accuracy" is highly improbable (but of course not impossible)."

Line 241-242"….. but the interpretation is hampered by the challenges of finding pristine and well-preserved corals and to the uncertainties related to the water depths above the corals." Considering the general sense of this short introduction in the conclusion the point of "well-preserved" is out of scope here.

Lines 243-245 "i) POS have the ability to define the discrete position (Fig. 3), hence,

they are considered sea level index points, whereas ii) SVS provide only an upper bound, and they are called limiting points (Fig. 4)."

I'm wondering if considering the nature of the special issue this discussion on the indicative meaning of the two archives can just mentioned before and this section starting directly with the discussion of the indicative meaning of POS.

Line 255 "….with absolute errors…." What do you mean precisely?

Line 315-329 I think that SVS are just terrestrial limiting point. So they cannot used to infer any special cases for the position of the RSL. Also hiatuses if not accompanied by clear evidences of marine deposition, for speleothems above sea level are dangerous to be considered as indication RSL changes. A more critical approach probably is necessary also considering the discussion you made at the beginning of the manuscript.

Moreover, here would be important to discuss a little also Yucatan Peninsula (which is mentioned as important at line 342)

Figure 4. SVS elevation indicating maximum positions of RSL during the time of their deposition. Not very clear SVS are terrestrial limiting point so the RSL is below them, and it is not always clear when they stop, it is an assumption that they stop just during flooding. They can stop also before. The top can experience dissolution and so on. Just a note of caution.

Considering Fig. 4 there also included not submerged speleothems now above sea level and (on the contrary some need to better explained to the reader). If it is the case probably some other papers are forget. For instance in the Mediterranean there is the recently published paper Bini et al. 2020 QSR but there are also others. So, I think probably there Maybe the original paper explain why these speleothems can be considered SVS, but this not emerge from the manuscript. Note that Wainer et al. stated: "The timing of growth of speleothems, at elevations close to sea level can provide records of minimum relative sea level (RSL) (note you often state maximum?).

In this study we used U–Th dating to precisely date growth periods of speleothems from Bermuda which were found close to modern-day sea level." This is a special prerequisite (even if in my opinion growth stop is not enough evidence of sea level changes. . ...). I think the reader looking at figure 4 maybe confused, so some more explanation is necessary.

Overall, I think the manuscript needs some clarification and expansion of the discussion in some points but careful (moderate) revision I hope will help improving the general quality and importance of this contribution.

---

## Referee Comment (RC3) · David Richards (Referee) · 20 Jan 2021

Last Interglacial (sensu lato, ∼130 to 75 ka) sea level history from cave deposits: a global standardized database

————

General comments:

Dumitru et al. have compiled relevant data from the submerged speleothem archive that can be used to provide valuable relative sea level data for the last interglacial period (130 – 75 ka). They focus on a specific, but loosely defined time window - the

last interglacial period- that will see plenty of attention because of the multitude of allied studies for this 'analogue' for future sea level change.

This component of the WALIS database will prove to be extremely useful. Speleothems provide robust constraints on sea level elevation, and phreatic overgrowths on speleothem provide precise sea level index points. The community needs such a framework; one that demands comprehensive documentation (and is sadly incomplete and inconsistent for some datasets – I am guilty myself for some of the gaps, and I have learned a great deal by revisiting the older data tables). Having said that, the database (and data) needs attention. The accompanying text also needs corrections and clarifications (see below).

I would like to see some indication of the management and governance of this facet/query of the database (i.e. speleothems). Some of this is inherited from WALIS general principles and guidelines, but it would benefit from repetition in brief, within the text. Is this a 'living dataset', and hence first version, or might it be described as an illustration of what might be call a thematic subset of data from the global standardised database WALIS?

On data coverage: Comprehensive, as far as I am aware, for MIS stage 5 in the strictest sense, but it would be worth reflecting on samples that have ages within the time window in title, but outside marine isotope stage 5 and could usefully be included because they help constrain the timing of the last interglacial (see data, for example, from the Bahamas incl. Gascoyne et al, 1984; Richards et al 1994; Smart et al, 1998 – see references).

A key concern (for authors and database managers, editors) that relates to this database (and SISALv2, Comas-Bru et al ., 2020) is the consistency of reporting measured U and Th ratios and terminology with respect to 'corrected ages'. Corrected ages as described in the database attributes are strictly 'initial Th-corrected ages' (using a correction based on a priori estimate of initial Th). 'Corrected ages'

are distinguished from 'recalculated ages', which are based on new information (improved decay constants and or further insight into initial Th). There is also the possibility of 3-d U-Th isochron ages that might require an alternative column (and not require 'correction' for detrital Th). Please address these differences in the text and consider alternative attributes. It would be worth commenting on the extent to which ages _could_ be recalculated for speleothems, perhaps by referral to the supplementary information at https://essd.copernicus.org/articles/12/2579/2020/essd-12-2579-2020-supplement.pdf.

Also important – and not covered: Speleothems provide periods of continuous growth and additional chronological information beyond the U-Th ages alone. To accurately constrain sea levels, one needs to know the growth rate, sample mid-point etc to define start and cessation of growth. This is a challenge to accommodate in the database, but ought to be considered or at least discussed in relation to the age-model work in SISALv2 etc.

**Please reinforce statements seen elsewhere in speleothem review articles related to the safety risks of sample collection in such settings and also encourage the adherence to principles of conservation and preservation of these caves. Returning to samples sites should be discouraged if sufficient material and appropriate documentation has been archived by the original authors. **

————

Details:

—

Abstract. Line13 on. Cave deposits include much more than the POS and SVS presented here, one can get beach deposits and corals within caves that provide sea level information. Declare focus on secondary carbonate precipitates, or speleothems. Best to be more specific.. many cave deposits are not suitable for U-series dating. There

are examples of cave sediments having been dated by OSL. Cave deposits can provide archives of valuable information, alternative to 'powerful archives'.

Line 16. POS on pre-existing vadose supports.. but what of walls? Title and elsewhere – explain the terms sensu stricto, lato (strictest sense, broadest sense) – not commonly used. And perhaps refer to alternative texts that define the last interglacial (marine isotope stage 5 or 5a-e or 5.1-5.5 etc)

Line 24. Focussed on MIS 5e, but also data that have the potential to constrain sea level fluctuations during the longer duration MIS 5. Not just 5a and 5c, because speleothems assist with constraining low stands also (e.g. 5b and 5d).

Line 26 use U, Th isotopes used to generate U-Th ages

Line 28. Why 'more importantly'? And (i) and (ii) need rewording.. One would expect abstract to address the 'living database' aspects.. unless you think this is a given because of the journal demands.

—

Section 1.

Line 38.. Expand on the 'samples and samples sites ..are better preserved'. I presume you mean that the context has been preserved and hence interpretation easier.

Line 38 'sea level evolution' – choose alternative.. and do you mean relative sea level

Line 40 'Yet to date . … are being debated' reword.

Line 46. There are many potential references, examples so use e.g Thompson et al., 2011.. etc. and consider Chutchavaran and Dutton (ESSD, this volume)

Line 50 use 'providing' rather than 'suggesting' maximum elevations

Line 55 include S+M 1972.. and maybe Dill, R.F., 1977, The blue holes, geologically significant submerged sinkholes and caves off British Honduras and Andros, Bahama

Islands: 3rd International Coral Reef Symposium, Proceedings, v. 2, p. 237–242. It was recognised 50 years ago that such deposits would be useful see Benjamin (1970) in National Geographic

Line 82. Age of carbonate material below hiatus is a _maximum_ age for timing of submergence (assuming no post-depositional alteration of exposed surfaces)

Line 83. Avoid the 'dipstick' analogy.. dipsticks indicate level, not binary status.

Line 84.. choose alternative to roughly.. constraints are robust and accurate (elevation), but there may be lag between the constraint and changing water levels.

Line 87 on. Use the term minimum estimate. I would challenge that initiation of growth can be rapid after emerence, but difficult to predict. The likelihood/duration of lagged response has yet to be established. Rough estimate is overstating things. Another advantage, not mentioned in the work is that further growth over the earliest calcite, post emergence, is 'protected' by subsequent growth. This is not the case for ages prior to hiatuses, which are susceptible to diagenetic alteration.

Line 94.. they do not document the "moment".. they can constrain the maximum and minimum age of sea level change at this depth.

Line 95. Time will tell.. speleothems have the potential to document the rise and fall of sea levels in stage 5. Maye the that ideal sites have not been identified yet.

Line 96. Choose alternative to 'difficult'. The data are unambiguous and easy to interpret, once it is established that they are constraints and NOT sea level index points.

Fig. 1. A useful figure, but needs refinement. e.g. constraints on sea level in (b).. label vadose calcite and hiatus/biogenic overgrowths. (c).. grey background, poor reproduction and use of T1 to T3 will have reader question why the internal morphology with changing MSL is not illustrated. POS should start earlier than T1 and at lower elevations than illustrated.

Line 116.. shorten to 'Cave deposits have received little or no attention in prior compilations'.

—

Section 2

Line 146 'varying' quality.. avoid. I'd accept 'variable'.. but you have not discussed criteria thus far. And on this ote, it would be worth illustrating the range in RSL data quality (1 to 5, compiler defined) For this section, MIS stage 5 boundaries might be determined by material that falls outside MIS 5 (e.g. Bahamas data in Li et al 1989; Richards et al, 1994; Lundberg and Ford 1994).

Table 1. Sites For 'not mentioned' (Moseley et al, 2013) – numerous caves in Quintana Roo, or similar. Additional information on Bahamas samples/sites. Richards et al., 1994. AN samples are from Stargate, Andros; GB samples Sagittarius Cave, Grand Bahama.

Line 160. Barometric altimeter (or diver depth gauge) adjusted for density profile, where possible with information about the vertical density profile associated with freshwater, brackish and saline zones. This is declared in database but referred to in discussed in text.

Line 177. Sample ID – samples may have been referred to in a number of papers. Declare best endeavour to find the first occurrence for sample ID might be useful phrase here.

Line 180. Reported ID - this might be sample ID or lab code. Database may need expanding to reflect this. For many geochronology laboratories, the analysis has separate codes to that provided by the group responsible for sub-sampling (e.g. Moseley et al 2013) .

Line 192. Please allow for possibility of using broken speleothems where possible, to preserve the aesthetic quality of caves. For many cases, the original location of a

broken sample can be established.

Line 194. Mineralogy does not affect the reliability.. it may dictate the susceptibility to alteration, but equally important is the geochemical setting. Low magnesium calcite is a relatively stable form of calcium carbonate in fresh and/or saline water, but mixing zone corrosion can cause serious dissolution. Encourage use of petrographic investigation. Also, marine borings, encrustations etc can dramatically alter the isotopic signal

Line 204 Dutton et al (2017) "is very applicable... for speleothems aswell". It was written to apply to all U-Th geochronology. Please reword.

Line 209 use 'alpha spectrometry', or 'alpha counting'. 'Alpha detector' is use in database, which is usually reserved for more basic instruments.

Line 211 use measured isotopic 'ratios and concentrations', not 'characteristics'.

Line 212. Use 'analyses of well-characterised internationally-recognised standards or certified reference materials are used to demonstrate the reliability of results'.

Line 213 'reduced' age uncertainties – not 'lowered'

Line 214 delete 'progressively'

Line 217 insert 'reported alpha spectrometry . . .or TIMS _results_'

Line 220 insert used: initial ratios and the decay constants used

Line 221 delete the effect of detrital thorium concentration. NB it is not only detrital Th one needs to consider, but also what some would call hydrogenous Th. Use of a threshold value value of 300 (Helstrom, 2006) for 230Th/232Th is arbitrary and depends on the ratio used for initial Th.

Line 230 on. This text needs to be improved. I think it is fair to accept that if ages are not presented as BP (or AD1950) or yr b2k, you can assume that dates are calculated with respect to date of analysis.

Line 232 declare ± 100 years (2 sigma) for age uncertainties for material of last inter-glacial age and add typical U concentration and declare insignificant 230Th initial

—

3. Discussions

Line 236. correlated to the _same_ (insert?)

Line 250 replace 'clear' with 'unambiguous'

Line 255 delete 'absolute', also declare early in text that all age uncertainties are quoted at 2 sigma.

Figure 3. You declare all ages are corrected for detrital Th. Begs the question – in original paper?, using updated decay constants? Please add more information.

Line 281. Calcite growth above and below (or before and after) a growth hiatus (not bottom and top).

Line 283. Again, avoid use of 'roughly'

Line 288 from not form; deciphering the timing of stillstands – if that is what you mean

Line 290.. be consistent you use mapsl elsewhere and m here. Is there a difference?

Para L300 on. Andros – in addition to the data for Gascoyne et al (1979), there are data in Gascoyne (1984), Richards et al (1994), Smart et al (1998) - see references above. Mostly alpha-spectrometric or TIMS U-Th ages.

Figure 4 does not include dates from Andros (Gascoyne, 1984; Richards et al 1994) or Grand Bahama (Richards et al, 1994; Smart et al, 1998). Not necessary, but there is a suggestion that data relating to MIS 5 in broadest sense is included. Also.. plotting U-Th ages alone only goes so far. The duration of continuous growth is required (see earlier comment). This figure does not illustrate this. It is not necessary, but the point needs to be made that these are ages only and _not_ growth periods. Awkward

placement of legend, it distracts from the data.

Line 365. The Bahama archipelago (includes the Bahamas and Turks and Caicos).

It is worth noting the work conducted by the Miami group to assess the subsidence of the Bahamas, in part because of loading through periodic carbonate production on the platform. e.g. McNeil, D.F.; Ginsburg, R.N.; Chang, S.-B.R., and Kirshvink, J.L., 1988. Magnetostratigraphic dating of shallow water carbonates from San Salvador, Bahamas. Geology , 16(1), 8–12.

In addition.. it would be worth acknowledging the additional constraints on sea levels that are provided by the flank-margin caves that host such speleothem deposits in some places. Perhaps at the same time as notches are mentioned (e.g. papers by Mylroie et al., 2020; Carew and Mylroie, 1990; Mylroie and Carew, 1995).

From journal guidelines: Ma and Myr (also Ga, ka; Gyr, kyr): "Ma" stands for "mega-annum" and literally means millions of years ago, thus referring to a specific time/date in the past as measured from now. In contrast, "Myr" stands for millions of years and is used in reference to duration (CSE, p. 398; North American commission on stratigraphic nomenclature).

————

The data compilation/spreadsheet (speleothems - last interglacial from WALIS):

It would be useful to refer to the site https://walis-help.readthedocs.io/en/latest/ (or acceptable zenodo URL) earlier in body of text. A lot of critical relevant material can be found here.

File name needs to be more specific with date, perhaps? Or is this the standard file name for a query download? I note that other database compilations have name of first author compiler and version/ date.

Tuccimei et al. URL link is broken

For data on vertical movements – what qualifies here? Needs guidance.

Please comment on the "Quality of age information", or signpost where this information can be found for speleothems in particular.

A component that needs to be considered is the definition of continuous growth. The age of growth initiation and cessation is dependent on sample position and growth rate. This caveat needs to be included in the paper. Would the database be expanded to include such information?

Reported age – please define (is this uncorrected for initial 230Th, for example). And what does corrected reported age mean? There is information at the WALIS website, but this should be included in the text for this paper.

Please reflect on the reporting of symmetrical U-Th age uncertainties for last interglacial ages. delta234U (declare that these are per mil values).

There is an assumption that [U] = [238U]..  e.g Dorale et al (2010) quote [U] ppm, but spreadsheet is [238U]*. Please comment.  *Only 0.73% difference, but outside uncertainty of typical measurement.

In database.. please distinguish between activity ratios and abundance ratios. Generally OK, but see heading for 230Th/232Th.. this is activity ratio.

Report latitude and longitude to full precision declared in paper ie. 87.00 rather than 87 to avoid confusion (see Moseley et al, 2013).

Explain the difference between coordinates and reported co-ordinates (latter not complete, see Cova del Dimoni, Mallorca).
* * *
Additional references referred to in text above

Mylroie, J.E. and Carew, J.L. (1990) The flank margin model for dissolution cave de-

velopment in carbonate platforms. Earth Surface Processes and Landforms, 15(5), 413–424.

Carew, J.L. and Mylroie, J.E. (1995). Quaternary tectonic stability of the Bahamian Archipelago: Evidence from fossil coral reefs and flank margin caves. Quaternary Science Reviews, 14(2), 144–153.

Mylroie, J, Lace, M., Albury, N and Mylroie, J. (2020) Flank Margin Caves and the Position of Mid- to Late Pleistocene Sea Level in the Bahamas. Journal of Coastal Research, 36(2), 249-260.

Gascoyne M. (1984) Uranium-series ages of speleothems from Bahamian blue holes and their significance. Cave Science, 11(1) 45-49.

Smart PL, Richards DA, Edwards RL. (1998) Uranium-series ages of speleothems from South Andros, Bahamas: Implications for Quaternary sea-level history and palaeoclimate. Cave and Karst Science 25(2), 67-74.
* * *

---

## Editor Comment (EC1) · Attila Demény (Editor) · 20 Jan 2021

I gratefully thank the reviewers for their careful and detailed reviews.

Attila Demény topical editor

---

## Short Comment (SC1) · 23 Jan 2021

As guest editor of the WALIS Special Issue, I would like join the handling editor in thanking the three reviewers for their constructive comments on this MS.

In particular, I take note of the comments by David Richards on the WALIS structure and data fields. As I am directly in charge of that, I will try to implement them wherever possible in the next version of the database (although I fear it will take some time), or to discuss them within the editorial article that will collate the SI papers (co-authored by the SI guest editors). Also, the readthedocs explaining the fields will be soon available on Github. This way, anyone will be able to suggest specific changes to field descriptors

in case they are not clear.

To answer the question about "live data", with the editorial that will close the SI we will make available WALIS v1.0, which will contain all the data inserted within each MS (in multiple formats, such as mySQL, CSV, and XLS), together with python notebooks to quickly extract data and a user-friendly interface for the quick visualization and download.

Alessio Rovere

---

## Author Comment (AC1) · 12 Mar 2021

Vanessa Johnston (Referee)

General comments:

The dataset compiles chronologically constrained, paleo relative sea-level based on previously published proxy records from cave deposits. The dataset has been compiled using a standardized format as part of a wider project to collate worldwide paleo sea-level data. In this way, the data are presented in a format that is easily usable by researchers who are not necessarily experts in cave deposits. This unique dataset, focusing on cave deposits, is an important addition to the database of more commonly used sea-level proxies, such as corals. The data, extracted from the original publications, appear to be inserted in a thorough and complete manner.
The manuscript accompanying the dataset is informative and well written. It states clearly how these deposits form and their significance to reconstructing paleo sea- level, with the use of simple diagrams, enabling researchers from other fields to under- stand the formation processes of these unique cave deposits. Moreover, it discusses the limitations of these deposits for assessing the paleo sea-level, both temporally and spatially, and outlines the uncertainties on these values with respect to the data in the dataset. The discussion presents the data in simple graphical forms and remarks on the significance of these data for paleo relative sea-level, glacial isostatic adjustment, and tectonic uplift. As would be expected from a manuscript accompanying a dataset, the authors have made no attempt to analyze the data in terms of paleoclimate/tectonics but have outlined some of the conclusions from the original publications and identified research directions to improve and expand the dataset.

*A: We appreciate your positive feedback and your suggestions. Below we address your comments point by point interspersing the review (in black) with our response (in green italics).*

Specific comments:
The POS are said to form in the brackish waters of the seawater–meteoric water mixing zone (line 61). In general, thermodynamic models of the mixing between seawater and calcite-saturated meteoric water produce aqueous solutions that are under-saturated with respect to calcite, and thus, promote calcite dissolution. The authors later note that the POS only form with the fluctuating water table, which is a key concept from a geochemical perspective and could be easily overlooked. I feel that this explanation of POS formation should be slightly expanded to overcome problems that may arise when a reader might expect calcite under-saturation in this zone, rather than the carbonate precipitation that is evident from the formation of these cave deposits.

*A: To address your concern regarding the POS precipitation and to clarify this process for the reader we added the following text (63-74):*

*"POS form on submerged cave walls and pre-existing vadose speleothems at and just below the water table (Ginés et al., 2012), when seawater mixes with meteoric water inside caves that are located in the close proximity to the coastline (within 300 m). The pre-existing vadose speleothems become partly submerged in the resulting brackish water. Previous petrographic investigations of these deposits suggested that the major control on carbonate precipitation is the ability of $CO_2$ to degas across the air-water interface (Pomar et al., 1976; Csoma et al.,*

*2006). These findings are supported by present-day observations that indicate the upper 40 cm of the water column being supersaturated with respect to calcium carbonate allowing for POS to form (Boop et al., 2014). While meteoric-marine mixing zones are mostly referred as sites of extensive dissolution, aragonite or calcite precipitation occurs when a high concentration gradient between $pCO_2$ of the cave water and atmosphere exists. Therefore, faster degassing is expected to happen in caves with low $pCO_2$ in their atmosphere. Corrosion of carbonate minerals was noticed in some Mallorcan caves, particularly when approaching the halocline; however, both calcite and aragonite are presently precipitating at the water table in the mixing zone, where numerical model predicts dissolution (Csoma et al., 2006)."*

Figure 1:This figure provides a good overview of the formation of POS and SVS deposits. However, I find that part b should have a better explanation to explain more clearly what the arrows are showing. In part c, the graph is very small and difficult to read in comparison with the other parts of the figure. Furthermore, it is a conceptual model but there are values on the y-axis; are these values referring to a specific case? Perhaps the x-axis also requires an arrow indicating "time". As a suggestion, the size of the orange oval (POS deposit) could be increased over time through points T1–T3 to show that it is growing. The inset photo in part c should probably include a scale and location information. Part d requires a scale.

*A: Thank you for your suggestions. Figure 1 has been updated accordingly.*

[Figure]

Technical comments:

Ln 13 and throughout: Please check consistency of text between "sea level" and "sea- level".
*A: We checked and used consistently "sea level" when used as noun and "sea-level" when used as an adjective.*

Ln 32: Remove the word "former" as it is not needed because you have "paleo" in the sentence. Also, sea-level should perhaps be pleural since the sentence ends with "them".
*A: We updated the text accordingly.*

Ln 36: Please add the word "period" after "the last interglacial".
*A: Added.*

Ln 40: Consider adding "partial", or similar, before the word "analog", as it is not a strict analog due to a different orbital configuration.
*A: Added.*

Ln 49: Please add the word "level" after the word "sea". Consider converting "still stand" to "stillstand" and, throughout the text, also consider the usage of "high stand or highstand", and be consistent.
*A: We updated the text accordingly.*

Ln 52: The letter "L" in the word "AnaLysis" should probably be capitalized following the SISAL acronym.
*A: Corrected.*

Ln 61: Consider changing "sea water" to "seawater" and check consistency throughout the manuscript.
*A: Changed and used consistently throughout the manuscript.*

Ln 119: Please add the word "period" after "the last interglacial".
*A: Added.*

Ln 133: Perhaps quote "xls" as "Microsoft Excel .xls" (or indeed .xlsx), or similar.
*A: Done.*

Ln 135: Please change the word "which" to "that" (because "which" should almost always be preceded by a comma).
*A: We updated the text accordingly.*

Table 1: This table is a bit difficult to follow since the locations/deposit types have been placed central to the row to which they refer. Please format in a different way.
*A: We agree and we formatted the table such that it is now easier for the reader to follow.*

Fig. 2 caption: Please change "modelled" to "modeled" as you are using American English.
*A: Changed.*

Ln 207: Please remove the word "times".
*A: We removed this word.*

Ln 262: Please consider changing the word "high" to "large" [uncertainties].
*A: Done.*

Ln 265: It is unclear to what the word "respectively" is referring to.
*A: The word "respectively" is not needed in this sentence and so, we deleted it.*

Fig. 3 caption: Remove the full stop after the word "Figure".
*A: Removed.*

Ln 284: Please insert parentheses around the "e.g.," clause. For example (e.g., your text here).
*A: Done.*

Ln 294: The dates quoted here require a citation in this sentence.
*A: We added the reference: Wainer et al., 2017.*

Ln 316: Please change the hyphen to an en-dash between "Atlantic" and "Caribbean".
*A: Done.*

Ln 318: Please remove the space between "5" and "c".
*A: Removed.*

Ln 319: Perhaps it would be clearer to write "reaching above −11 m, but not as high as −4.9 m" as "between −11 m and −4.9 m".
*A: We agree and we changed the text accordingly.*

Fig. 4: Change Harmon citation date from 1878 to 1978!
*A: Changed.*

Ln 347: The word "authors" should be authors' (add apostrophe) because it is a possessive meaning being used.
*A: Done.*

Ln 348: Please remove the words "the former".
*A: We removed this word.*

Ln 363: Please use an en-dash between the "5c" and "a" (i.e., 5c–a).
*A: Done.*

Ln 385: Please change "refining" to "to refine".
*A: Done.*

Ln 401: It is unclear to which island you are referring (Mallorca or Sardinia).
*A: We clarified in the manuscript and added Mallorca.*

Ln 402: Please remove spaces either side of the en-dash.
*A: Done.*

Ln 403: Please use an en-dash, rather than a hyphen, between the words "Pleistocene" and "Holocene".
*A: Done.*

Ln 407: Please use an en-dash between the values 600 and 400 ka.
*A: Done.*

Ln 408: Please change hyphen for an en-dash and remove spaces before and after to become "MIS 11–MIS 5e".
*A: Done.*

Ln 415: Please add the word "period" after "the last interglacial".
*A: Added.*

Ln 424: Please consider removing the "etc." and changing the sentence to a phrase, such as "... benefit research related to other disciplines, such as, water resources availability, sea-level rise, and saltwater intrusions."
*A: Done.*

Ln 430: This line states "elevations much lower than present in caves", I presume you are referring to present sea-level, please then add the word "sea-level" into this phrase.
*A: Added.*

---

## Author Comment (AC2) · 12 Mar 2021

I have now completed my review of the manuscript of Dimitru et al. I think it is an useful contribution with several interesting considerations, which would be further expanded in the future works. Considering I'm not an expert of the WALIS database I don't give feedback on the database itself even if the most relevant information seem to have been considered. I have several points along the text which I can summarise in some general comments:

- From the text there is no a clear discriminant petrographic, geochemical and morphological criteria to separate POS from other speleothems which forms in freshwater pools in caves. I think this would be very useful to give this description to clarify that these are unique speleothems typology.
*A: We agree that this problem could be a concern and should be more extensively discussed. Nevertheless, there are only a handful of such studies on POS, mainly because their occurrence in coastal caves at distinct elevations clearly relate them with present and past still sea level stands. In order to highlight the morphological, geochemical and petrographic particularities of POS, we included in the manuscript the following text starting with line 80:*

*"With very few exceptions, the morphology of the POS is clearly different from that of speleothems precipitated at the fresh water level in pools from non-coastal caves, e.g., shelfstones and subaqueous freshwater pool spar, on which the overgrowths are truncated in the upper part and mainly accrete under the water level. Furthermore, the carbonate deposition of these speleothems is not symmetric with respect to the water level and the tide range, which is a particularity of POS. The only instance when POS form just under the water level is when the preexisting vadose speleothem (i.e., stalagmite) was not long enough to capture the full range of the tide, which is responsible for their spherical or elliptical morphology. POS can take a variety of shapes and sizes, depending on the morphology of the vadose support, for how long they were immersed in the cave's brackish water, and the tide amplitude. Only few petrological and geochemical studies have been performed so far (Pomar et al., 1976; Ginés et al., 2005, 2012; Csoma et al., 2006). The mineralogical and crystallographical data indicate calcite as the dominant phase with fibrous, elongated, and isometric crystals, but radial-fibrous/acicular aragonitic fabric can exceed 70 % in some samples (Ginés et al., 2012). A limited number of stable isotopes analyses showed an isotopic evolution towards heavier composition through the MIS 5a and 5e possibly due to excessive marine water intrusion in the cave ponds (Vesica et al. 2000). More in-depth studies have been undertaken to investigate the POS precipitation conditions and the relationships between surface conditions (temperature, barometric pressure, precipitation, tidal level of the sea) and the microenvironment of coastal caves (temperature, partial pressure of $CO_2$, and water level; Boop et al., 2014). The distinction between POS and shelfstones (flat deposits attached to cave walls or on partly immersed speleothems that grow inwards from the edge of the pool/speleothem) is clearly described in Onac et al. (2012)."*

- SVS are basically terrestrial limiting points. The use of hiatus as indicator of sea level submersion in absence of other clear evidences is probably dangerous and risk to be overexploited (hiatuses form in speleothems form many reasons). This is clearly stated in the manuscript. But then is not considered anymore as criteria and probably a more critical approach for previous data should be considered.
*A: The reviewer is absolutely right in that SVS are terrestrial limiting points. Various approaches were considered in tackling the presence of hiatuses, which as the reviewer*

*mentioned, could be due to other climatic and hydrologic factors. Detailed petrographic studies can confidently assess what caused a particular growth hiatus, especially if a speleothem is submerged. During the inundation, the following characteristic features may form: i) the dissolution at the halocline produces corroded layers; ii) biogenic encrustations; iii) traces of marine borings; and iv) deposition of various trace elements or minerals (halite, gypsum, etc.). The carbonate layer just below the hiatus will provide a maximum age estimate for when this location in the cave was air-filled and sea level was positioned below the speleothem elevation. For better understanding the limitations of the SVS as sea-level indicators we rewrote the introductory paragraph from Submerged vadose speleothems section from line 192-206:*

*"Speleothems such as stalactites and stalagmites form in air-filled passages, thus their periods of growth indicate times when sea level was lower; hence they are sea-level terrestrial limiting points. For vadose speleothems that are subject to sea-level submergence, hiatuses (i.e., no carbonate deposition) can be correlated with periods when sea level rose and inundated the cave causing speleothem growth to cease. However, speleothem growth cessation is not always related to sea level rise as other climatic and hydrologic factors could stop carbonate precipitation. When growth cessation is sea-level related, particular mineralogical and/or biological features can be visible using petrography. Some of these include: i) corroded layers when dissolution happens at the halocline; ii) biogenic encrustations; iii) traces of marine borings; and iv) deposition of various trace elements or minerals (halite, gypsum, etc.). Details regarding ways of deciphering different types of growth hiatuses are presented by Onac et al. (2012) and van Hengstum et al. (2015). Dating the carbonate layer immediately above each of these hiatuses provides a minimum estimate of when the cave became air-filled again constraining the minimum age for the sea-level fall. The carbonate layer below a hiatus indicates the maximum age, assuming no post-depositional alteration of the exposed surfaces, for when this location in the cave was air-filled and the sea level was clearly below the speleothem elevation. It is worth noting that the earliest layers deposited above the hiatuses are protected by further carbonate precipitation, whereas those below the hiatuses are susceptible to diagenetic alteration."*

- Detrital correction in speleothem dating is critical. Probably a short mention on the fact that different laboratories applies different correction is useful (also in the past). It would be useful to have a range of 232Th/230Th values found is POS speleothems for the reader in the text.

*A: We included this information in the revised manuscript and the text now reads as follows (lines 283-297):*

*"The correction for the initial non-radiogenic sources (i.e., hydrogenous, colloidal and carbonate or other detrital components; Richards et al., 2012) of $^{230}Th$ incorporated at the time of speleothem deposition is extremely important for age calculation and is sensitive for samples that contain very little uranium or an abundance of detrital thorium. The $^{230}Th/^{232}Th$ activity ratio of 0.825 with an arbitrarily assigned uncertainty of 50% found in the mean bulk Earth or upper continental crustal has been commonly assumed for initial $^{230}Th$ corrections. However, several studies have shown that this value may not cover all situations. Therefore, laboratories apply different corrections for the non-radiogenic detrital $^{230}Th$ fraction through either direct measurement of sediments associated with speleothems (Hoffmann et al., 2018) or computed*

*isochron methods and stratigraphical constraints (Hellstrom, 2006; Richards et al., 2012). Most POS included in this database fulfill the criterion suggested by Hellstrom (2006) that samples with ratios of $^{230}Th/^{232}Th$ higher than 300 are considered clean, with very few samples having lower such ratio values (Tuccimei et al., 2007). However, the use of this threshold value is arbitrary and depends on the ratio used for initial Th."*

- Along the text there are many sentences unclear and/or not very precise (in my opinion). I have suggested modification. Some sentences need to be supported by references.
*A: In the revised manuscript we addressed your comments and suggestions and included the requested references.*

Overall the manuscript is easy to read (even if not all sentences are consequential) and general conclusions interesting even if probably (considering the nature of the special issue and that of the manuscript) a section including some general more methodological approach in selecting material and advices as use data for some conclusion would be useful.
*A: We thank you for your suggestion. We consider including in the manuscript the following section with the necessary information that we strongly encourage researchers publishing new sea-level studies based on speleothems to report.*

*"To build a more valuable dataset that will have more longevity and use within the discipline, we strongly encourage researchers publishing new sea-level studies based on speleothems to include the following information:*

- *Sea-level indicator and its relationship to sea level: i) site location (latitude and longitude of the cave); ii) the elevation of the sea-level indicator, the instrument type used and its precision, and the error associated with the elevation measurement (when using barometric altimeter or diver depth gauge for submerged samples, the elevations should be adjusted for density profile); iii) the sea level datum to which the elevations are referred and how the indicative meaning has been quantified.*
- *Screening results: XRD, petrography, and polarizing/scanning electron imaging, including information on mineral assemblage, as well as diagenetic and crystallization descriptions (e.g., fabric).*
- *U-series data: In order to collectively improve the utility of U-series data we encourage researchers publishing new sea-level studies based on speleothems to follow the recommendations suggested by Dutton et al. (2017) in reporting their data. These authors specify the required data to enable calculation and, if needed, re-calculation of the same ages using different parameters and also, to facilitate the interpretation in the context of other studies. The checklist of minimum data to report includes: uncertainties for all parameters, state whether uncertainties on ages include decay constant uncertainties; Names, descriptions, and reference values of reference materials; Decay constants; Isotopes in spike and method of spike calibration; Method of calibration for all activity or atom ratios reported; Activity or atom ratios for $^{230}Th/^{238}U$ (or $^{230}Th/^{234}U$) and $^{234}U/^{238}U$; $^{230}Th/^{232}Th$ activity or atom ratio; Details of procedures and values used to interpret ages using isochrons or other models; Date of analysis or reference age (e.g., BP, b2k, etc.). These recommendations will increase the usefulness of this type of analytical results in the U-series geochronology community (Dutton et al., 2017). We*

*also recommend reporting continuous growth rate which allows to better define the onset and cessation of deposition for either POS or SVS samples.*

Specific comments (generally minor)

Line 40 ".... (Capron et al., 2019), and MIS 5e is considered an analog for the Holocene." The Holocene is quite different in terms of insolation and sea-level history. Please avoid oversimplification. MIS5e is not a good analogue for the Holocene, but it is the closest interglacial we can study with relatively good details. The main point is that MIS5e has temperature higher than pre-industrial Holocene, but other boundary conditions are very different.

*A: We agree with your comment and the text now reads: "MIS 5e is considered a potential analog for the future sea-level rise due to anthropogenic global warming since temperatures were on average ~ 1.5 °C higher than today (relative to the AD 1961–1990 period; Turney and Jones, 2010)."*

Line 43 ". . .uncertainties in the reconstructed sea level." Please Insert a reference
Lines 42-43 "Fossil corals can be dated to relatively high precision but have meter scale uncertainties in the reconstructed sea level" Please insert references

*A: We cited two of the main global compilations of corals sea-level indicators for MIS 5e: Hibbert et al., 2016 and Chutcharavan and Dutton, 2021, where the meter-scale constrained paleowater depth uncertainties are described in detail.*

Lines 43-44 "Other indicators such as erosional notches pinpoint sea level, but lack age control". Please adds citation Are you sure you want refer to erosional notches (they are not the best, but in case I can suggest Bini et al. 2014 Earth Surface Processes and Landforms, 39 (11),1550-1558). Note that this is the same problem for tidal notches, probably some like erosional sea level markers are difficult to be dated? Or? Some more general?

*A: We agree that notches are not the best sea level indicators and we had mentioned in our text that they lack age control. Thank you for suggesting Bini et al. (2014). This reference and Antonioli et al. (2015) are now both included in the revised manuscript as examples of tidal notches being used to estimate sea level position.*

Lines 44-45 For this reason, there is a growing demand in exploring additional sea level indicators that can complement the information derived from fossil corals, while simultaneously having robust chronology. Maybe the sentence is not completely true most of the indicators are well known but the point is to clarify the indicative meaning and to date correctly them.

*A: The text was revised to read: "For this reason, there is a growing demand in exploring sea-level indicators that can simultaneously provide a robust chronology and a clear indicative meaning."*

Line 46 ". . .as coral reefs (Thompson et al., 2011). . ." are these unique of karst environment?

*A: Thank you for noting this mistake, as coral reefs are not necessarily associated with karst environments and consequently, we removed them from the list of sea level indicators specific to karst environments.*

Line 50 ". . ..submerged vadose speleothems (SVS; suggesting maximum elevations of sea level position)." Not very clear. Why they should be maximum elevation? They indicate the cave inundation (see your line 80) or air filling, so they are terrestrial limiting points. I think we must be conservative in these concept or be more clear in the explanation I have the same observation to other points

*A: The confusion comes from the fact that we have not included a more thorough discussion regarding the meaning of hiatuses in SVS. This has been now clearly explained in Section 1.1. - "The relationship between SVS deposition and sea-level changes", where we included the concept of "terrestrial limiting points" and also a brief explanation for the relationship between carbonate layers below and above a given hiatus in SVS as presented at your previous comment (lines 192-206).*

Line 95 " was actually located throughout the bulk of the rise-fall cycle (Richards et al., 1994; Suric ́ et al., 2009). Therefore, it can be difficult interpreting the relationship between vadose speleothems growth and sea-level history." Yes, I agree. Translating in a few word they are terrestrial limiting points? Is that you want to say? Probably this concept can be introduced here considering the general nature of the special issue, to avoid to return to the same concept later. I think this point should highlighter later in the conclusion or in a special section where it should be stated that use of SVS hiatus need to be considered with cautions. Maybe showing some examples.

*A: We included the concept of terrestrial limiting points at your previous suggestion. We also added the following text in the Conclusions section:*

*Line 502-503: "This dataset paper represents the first compilation of cave deposits (POS as sea level index points and SVS as terrestrial limiting points) used to reconstruct sea-level histories for the last interglacial period."*

*Line 508-511: "Intervals of SVS growth is indicative of times when sea level must have been lower than their elevation. Hiatuses can also be used to indicate sea level position, however, they must be cautiously interpreted since there are several reasons which can lead to their occurrence. Hiatuses associated with biogenic encrustations or borings could potentially indicate more exactly where sea level was with respect to the SVS."*

Lines 166-170 "Several tools with different uncertainties have been used to measure the elevation of the cave deposits: barometric altimeter ($\pm$ 0.1 m; Moseley et al., 2013), metered tape or rod ($\pm$ 0.5 m; Harmon et al., 1978), inclinometer ($\pm$ 170 0.05 m; Dorale et., 2010)." I think here there is some confusion between instrumental precision and accuracy of the measure. Not clear. For instance barometric altimeter can give you this kind of precision, but it is quite far from the accuracy of the elevation measurement. All of this have been reported to local datum? Just to clarify.

*A: The uncertainties cited in the paper are the errors in measurements as reported by the authors in their respective studies. Yes, all samples have been reported relative to present mean sea level and we included this information in the revised manuscript in line 216.*

Line 175 ". . ..or that precise measurements are not so relevant because the uncertainties related to local tectonics are much larger (Suricʹ et al., 2009)." Sentence unclear precise measurements are mandatory. Maybe delete?

*A: We agree that precise measurements are mandatory, but nonetheless, the meaning of the sentence was just to emphasize that sometimes the corrections (e.g., tectonic uplift, GIA) are so much larger that the elevation uncertainty becomes less relevant when estimating a past sea level position. For more clarity, we updated the text to (line 223): "...or that the uncertainties related to local tectonics are so much larger that the measurement uncertainty (Surić et al., 2009).*

Section 2.5 "Given that speleothems are less likely subjected to alteration and diagenesis when compared with organic precipitation of corals, an elaborate sample pre- screening is not critical. Thus, only some of the studies compiled here report the mineral assemblage of the samples by X-ray diffraction (Suricʹ et al., 2009; De Waele et al., 2018). However, we cannot exclude the possibility that screening was performed, but not reported."
This is misleading section. XRD are fundamental in corals because they are aragonitic and tend to transform in calcite. Most speleothems are not aragonitic but calcitic. However, you stated at line 420 (why not before) that "The phreatic overgrowth mechanism that deposits calcite/aragonite at sea level. . .." So the problem for POS is real, and aragonite risk to be recrystalised all the same. However, it is a fundamental practice in speleothem science to select samples after a screening in thin section because also calcite can experience open-system behaviour. So this sentence should clearly state that (especially for POV and SVS) a check for potential alteration is necessary. A good case is the discussion made by De Waale et al 2017,2018, but for vadose speleothems there are many useful references for checking potential alteration. I think this section should be rewritten. I think what is lacking here which are the criteria to cite that POV are really POV and not formed in a lacustrine environment. I think there are not a very precise description here to discriminate them and to charaterised better POV.
Line 201-202 "While not always necessary, including information on mineral assemblage, and diagenetic and crystalline descriptions are useful." I think this is a mistake and I don't' think is the though of the writers. Petrographic and diagenetic description is fundamental and mandatory! There is no mention here on the clastic contamination (but it is later). Presumably in POV would be minor but according the compilation proposed it would be nice to have a range here of measured 232Th/230Th and eventually to give a range of correction performed by different lavoratories. This can have an effect on the final age.

*A: We agree and we rewrote this paragraph and it reads now as follows (lines 255-267):*

*"The geochemical setting and sample mineralogy may dictate the susceptibility to alteration. For example, samples that show conversion of aragonite to calcite or calcite recrystallization, could have been subjected to uranium loss, which is an important factor that impacts U-Th ages (Lachniet et al., 2012; Bajo et al., 2016). To allow recognition of diagenetic fabrics, XRD screening is desirable for dating purposes (aragonite is preferred vs calcite). In order to make the best selection of samples for successful dating, we encourage the use of petrographic investigation as well. Thin sections can help to best identify the speleothem layers just below and above the hiatus for dating purpose, and they also reveal the internal structure, hence areas affected by recrystallization can be avoided for dating. However, only some of the samples*

*compiled here have their mineralogy documented by X-ray diffraction (Surić et al., 2009; De Waele et al., 2018). De Waele et al. (2017), for example, complemented the screening method with petrographic investigations (thin sections) and imaging using scanning electron microscopy. We do not exclude the possibility that screening was performed in the other publications, but it has been not reported. For future studies, we recommend adding information on mineral assemblage, as well as diagenetic and crystalline descriptions."*

*We addressed the differences between POS and speleothems formed in cave pools as well as the detrital Th correction in our response to your comment above.*

Line 215 "The development of TIMS and then MC-ICP MS in measuring U-series isotopes constituted a major step forward from the alpha spectrometric method." Maybe here a reference is necessary.
*A: We added the reference: Hoffmann et al. (2007).*

Line 223-224 "Hellstrom (2006) suggested that a ratio of 230Th/232Th larger than 300 can be considered as an indicator of clean samples not requiring correction for detrital thorium." It is probably useful to mention that different laboratories applies different correction (e.g. Bulk upper crust or iterative calculation of the initial ratio) (see also previous point).
*A: We already addressed this concern in response to one of your previous comments.*

Line 232 ". . ...age errors are now possible to ± 100 years. . .," yes, but this kind of error for speleothems is probably just an analytical error. Considering clastic contamination and growth rate to have such a high "accuracy" is highly improbable (but of course not impossible)."
*A: We have now completed this sentence and it now reads (line 307):*

*"However, with the improvements made using MC-ICP MS (Cheng et al., 2013), ages on high quality samples (i.e., aragonite mineralogy, high U content, insignificant $^{230}$Th correction) of last interglacial are now possible to uncertainties of ± 100 years (2σ), making how the age is reported more important (i.e., BP)."*

Line 241-242". . ... but the interpretation is hampered by the challenges of finding pristine and well-preserved corals and to the uncertainties related to the water depths above the corals." Considering the general sense of this short introduction in the conclusion the point of "well-preserved" is out of scope here.
*A: Not necessarily. A well-preserved coral will return a very precise age and only **if** the depth of the sample can be estimated with a good approximation, sea-level position could be reasonably reconstructed.*

Lines 243-245 "i) POS have the ability to define the discrete position (Fig. 3), hence, they are considered sea level index points, whereas ii) SVS provide only an upper bound, and they are called limiting points (Fig. 4)." I'm wondering if considering the nature of the special issue this discussion on the indicative meaning of the two archives can just mentioned before and this section starting directly with the discussion of the indicative meaning of POS.
*A: We agree. We define POS as sea level index points and SVS as terrestrial limiting points in the introduction section, hence, we avoid using here redundant information.*

Line 255 ". . ..with absolute errors. . .." What do you mean precisely?
*A: For clarity, we replaced "absolute errors" with "ages uncertainties".*

Line 315-329 I think that SVS are just terrestrial limiting point. So they cannot used to infer any special cases for the position of the RSL. Also hiatuses if not accompanied by clear evidences of marine deposition, for speleothems above sea level are dangerous to be considered as indication RSL changes. A more critical approach probably is necessary also considering the discussion you made at the beginning of the manuscript. Moreover, here would be important to discuss a little also Yucatan Peninsula (which is mentioned as important at line 342)
*A: The ways in which a SVS could be used in reconstructing sea level position is now discussed in more details in Section 1.1, where we showed that hiatuses are not only caused by sea-level rise. Nevertheless, some features (biologic encrustations, borings) associated with hiatuses in SVS are clearly indicative of sea level. How precisely these are, is a different issue. Yet, for areas where no other RSL indicators exist, SVS could be used to at least estimate the position of past sea levels.*

Figure 4. SVS elevation indicating maximum positions of RSL during the time of their deposition. Not very clear SVS are terrestrial limiting point so the RSL is below them, and it is not always clear when they stop, it is an assumption that they stop just during flooding. They can stop also before. The top can experience dissolution and so on. Just a note of caution. Considering Fig. 4 there also included not submerged speleothems now above sea level and (on the contrary some need to better explained to the reader). If it is the case probably some other papers are forget. For instance in the Mediterranean there is the recently published paper Bini et al. 2020 QSR but there are also others. So, I think probably there Maybe the original paper explain why these speleothems can be considered SVS, but this not emerge from the manuscript. Note that Wainer et al. stated: "The timing of growth of speleothems, at elevations close to sea level can provide records of minimum relative sea level (RSL) (note you often state maximum?).
*A: We updated Figure 4 and its caption which now reads: "SVS (terrestrial limiting points) sample elevations and their U-series ages indicating RSL below (down-pointing triangles) or above (up-pointing triangles) them. Note that these are ages only and not growth periods. None of the data are corrected for GIA or long-term deformation."*

[Figure]

*We also included references that we overlooked in our first draft. As we explained in our answers above, the intervals of speleothems growth are indicative of maximum sea levels. It is unclear to us why Wainer et al. associate the duration of the hiatuses with minimum RSL in their abstract. The same authors state later in their text that: "the duration of the hiatuses requires a **minimum** RSL above the elevation of the ceiling from which the stalactite grew".*

In this study we used U–Th dating to precisely date growth periods of speleothems from Bermuda which were found close to modern-day sea level." This is a special prerequisite (even if in my opinion growth stop is not enough evidence of sea level changes. . ...). I think the reader looking at figure 4 maybe confused, so some more explanation is necessary.
*A: The reviewer is correct in assuming that not all speleothem growth is due to sea level rise. However, in coastal caves subjected to periodic flooding this is the main reason. We updated Figure 4 and its caption which now clarifies that RSL must be below the samples marked by the down-pointing triangles and above the ones marked by up-pointing triangles.*

Overall, I think the manuscript needs some clarification and expansion of the discussion in some points but careful (moderate) revision I hope will help improving the general quality and importance of this contribution.
*A: Thank you for your useful suggestions which have been very helpful in improving the manuscript.*

---

## Author Comment (AC3) · 12 Mar 2021

General comments:

Dumitru et al. have compiled relevant data from the submerged speleothem archive that can be used to provide valuable relative sea level data for the last interglacial period (130 – 75 ka). They focus on a specific, but loosely defined time window – the last interglacial period- that will see plenty of attention because of the multitude of allied studies for this 'analogue 'for future sea level change. This component of the WALIS database will prove to be extremely useful. Speleothems provide robust constraints on sea level elevation, and phreatic overgrowths on speleothem provide precise sea level index points. The community needs such a framework; one that demands comprehensive documentation (and is sadly incomplete and inconsistent for some datasets – I am guilty myself for some of the gaps, and I have learned a great deal by revisiting the older data tables). Having said that, the database (and data) needs attention. The accompanying text also needs corrections and clarifications (see below).

*A: We are grateful for your thorough review and very useful suggestions. Thank you also for catching some of the errors that added confusion into our text; these have now been corrected. Please find below a point-by-point answer to your comments.*

I would like to see some indication of the management and governance of this facet/query of the database (i.e. speleothems). Some of this is inherited from WALIS general principles and guidelines, but it would benefit from repetition in brief, within the text. Is this a 'living dataset', and hence first version, or might it be described as an illustration of what might be call a thematic subset of data from the global standardised database WALIS?

*A: Thank you for pointing out this missing information. We have now included in the revised manuscript the following text (starting at line 33 and at line 174):*

*"We refer the readers to the official documentation of the WALIS database at: https://walis-help.readthedocs.io/en/latest/ where the meaning of each field is thoroughly explained. All the data inserted in the manuscript will be available in WALIS v1.0, which will provide a user-friendly interface for quick visualization, extraction and downloading of the data."*

*For more details on the WALIS structure and data fields, please see the answer provided by Alessio Rovere, the editor who is directly in charge of taking care of this.*

On data coverage: Comprehensive, as far as I am aware, for MIS stage 5 in the strictest sense, but it would be worth reflecting on samples that have ages within the time window in title, but outside marine isotope stage 5 and could usefully be included because they help constrain the timing of the last interglacial (see data, for example, from the Bahamas incl. Gascoyne et al, 1984; Richards et al 1994; Smart et al, 1998 – see references).

*A: Thank you for suggesting these references we overlooked in the first draft of the database. We fully agree that data older and younger than MIS 5 would help constraining the timing of last interglacial. This is why we initially chose the broadest sense of MIS 5 from 130 to75 ka. We have now further extended this interval from 140 to 70 ka. However, to be in accordance with the time interval targeted by WALIS, our discussion does not include samples of ages outside the range mentioned above. Therefore, for completeness, the database will contain all ages reported in the original publications, but in the discussion of the manuscript we include only those between 140 to 70 ka. An updated version of the database will be uploaded shortly.*

A key concern (for authors and database managers, editors) that relates to this database (and SISALv2, Comas-Bru et al., 2020) is the consistency of reporting measured U and Th ratios and terminology with respect to 'corrected ages'. Corrected ages as described in the database attributes are strictly 'initial Th-corrected ages '(using a correction based on a priori estimate of initial Th). 'Corrected ages 'are distinguished from 'recalculated ages', which are based on new information (improved decay constants and or further insight into initial Th). There is also the possibility of 3-d U-Th isochron ages that might require an alternative column (and not require 'correction 'for detrital Th). Please address these differences in the text and consider alternative attributes. It would be worth commenting on the extent to which ages could be recalculated for speleothems, perhaps by referral to the supplementary information at
https://essd.copernicus.org/articles/12/2579/2020/essd-12- 2579-2020-supplement.pdf.
*A: We thank you for bringing this concern to our attention. To avoid any confusion, we now included in the manuscript the following clarifying text (line 293):*

*"We emphasize that this database includes the ages as they are reported in the original publications and therefore, it contains two columns: reported ages and corrected ages. The latter one represents the ages to which the original authors applied a correction for detrital Th based on a priori estimation of initial Th. We did not apply any further corrections when compiling this database, so it does not include any "recalculated ages". All data contains the minimum required information (as recommended by Comas-Bru et al., 2020, suppl.) to calculate uncorrected ages, however, not all of them provide the initial $^{230}Th/^{232}Th$ activity ratio to allow the calculation of detritus corrected ages."*

Also important – and not covered: Speleothems provide periods of continuous growth and additional chronological information beyond the U-Th ages alone. To accurately constrain sea levels, one needs to know the growth rate, sample mid-point etc to define start and cessation of growth. This is a challenge to accommodate in the database, but ought to be considered or at least discussed in relation to the age-model work in SISALv2 etc.
*A: We agree with the reviewer's point that speleothems offer much more than just the U-Th chronology, but for SVS, even if the periods of continuous growth (and rate of growth) are precisely documented, we cannot really tell where exactly the position of sea level was with respect to the speleothem. Nevertheless, acknowledging the importance of the reviewer's point, we added the following text in the revised manuscript (lines 137-145):*

*"The age of growth initiation and cessation is dependent on sample position and growth rate. A highly resolved U-Th chronology defines the degree of continuous growth and a high-quality petrographic examination of the sample would support that. These ages can be used to calculate the growth rate, allowing to better define the onset and cessation of deposition for either POS or SVS samples. This information bears significance since one can use the growth rate to project the onset of a hiatus, which in coastal caves provides evidence for when sea level emerged above that particular speleothem elevation. To provide robust chronologies with temporal uncertainties, we refer the readers to the workflow to treat records with hiatuses developed by Comas-Bru et al (2020)."*

**Please reinforce statements seen elsewhere in speleothem review articles related to the safety risks of sample collection in such settings and also encourage the adherence to principles of conservation and preservation of these caves. Returning to samples sites should be discouraged if sufficient material and appropriate documentation has been archived by the original authors. **
*A: Thank you for noting this important aspect that has been overlooked. We fully agree that this concern should be explicitly stated in the manuscript. We now included the following information in the revised manuscript (lines 245-252):*

*"The caves have unique scientific, recreational, and scenic value; hence, speleothems sampling strategies must be selective and reconcile cave conservation with the scientific goal (Baeza et al., 2018). For this reason, we emphasize that the utmost effort should be made to minimize the future impacts of sampling by following the conservation and preservation guidelines. We strongly encourage that great care must be paid to more sustainable sampling strategies and, if sufficient material and appropriate documentation has been archived by the original authors, making them available for future researchers is desirable. Similarly, we recommend to preferentially sample already broken speleothems whenever the original location can be established, as suggested by Frappier (2008). Another sustainable sampling strategy is coring the central part of speleothems rather than collecting the entire specimen and patching the drill holes (Spötl & Mattey, 2012)."*

Details:
Abstract. Line13 on. Cave deposits include much more than the POS and SVS presented here, one can get beach deposits and corals within caves that provide sea level information. Declare focus on secondary carbonate precipitates, or speleothems. Best to be more specific.. many cave deposits are not suitable for U-series dating. There are examples of cave sediments having been dated by OSL.
*A: We changed the text to: "Two main categories of secondary carbonate precipitates in the caves…"*

Cave deposits can provide archives of valuable information, alternative to 'powerful archives'.
*A: We updated the text accordingly.*

Line 16. POS on pre-existing vadose supports.. but what of walls?
*A: We adjusted the text to: "…precipitate on preexisting supports (vadose speleothems or cave walls)".*

Title and elsewhere – explain the terms sensu stricto, lato (strictest sense, broadest sense) – not commonly used. And perhaps refer to alternative texts that define the last interglacial (marine isotope stage 5 or 5a-e or 5.1-5.5 etc)
*A: We have now explained in the abstract: "Here we describe a compilation that summarizes the current knowledge of the complete last interglacial (in its broadest sense - sensu lato - also known as marine isotope stage (MIS) 5) sea level captured by speleothems". We also added in the Introduction: "Understanding sea-level changes during the last interglacial period (MIS 5e; in its strictest sense - sensu stricto - from 130–116 ky)."*

Line 24. Focussed on MIS 5e, but also data that have the potential to constrain sea level fluctuations during the longer duration MIS 5. Not just 5a and 5c, because speleothems assist with constraining low stands also (e.g. 5b and 5d).
*A: Included now in the text.*

Line 26 use U, Th isotopes used to generate U-Th ages
*A: This has been replaced in the manuscript.*

Line 28. Why 'more importantly'? And (i) and (ii) need rewording.. One would expect abstract to address the 'living database 'aspects.. unless you think this is a given because of the journal demands.
*A: We rephrased the text which now reads: "Furthermore, the paper emphasizes the usefulness of these indicators not only to render information regarding the eustatic sea level, but also for their contribution to refine the glacial isostatic adjustments models and to constrain regional tectonic uplift rates". We also included in the abstract the following: "We refer the readers to the official documentation of the WALIS database at: https://walis-help.readthedocs.io/en/latest/, where the meaning of each field is explained in detail".*

__ Section 1.
Line 38.. Expand on the 'samples and samples sites ..are better preserved'. I presume you mean that the context has been preserved and hence interpretation easier. Line 38 'sea level evolution ' –choose alternative.. and do you mean relative sea level
*A: We only meant to say that the records that formed during the MIS 5e are better preserved than those from older interglacial periods. For clarity, we rephrased to:" Sea level indicators that formed during MIS 5e are often better preserved compared to those formed in earlier interglacial periods and thus, the relative sea level during this time interval is especially informative".*

Line 40 'Yet to date . . .. are being debated 'reword.
*A: We rephrased to: "However, significant uncertainties regarding the precise timing, duration, and amplitude of MIS 5e sea level remain".*

Line 46. There are many potential references, examples so use e.g Thompson et al., 2011.. etc. and consider Chutchavaran and Dutton (ESSD, this volume)
*A: We cited in the updated manuscript: Hibbert et al. (2016), Chutcharavan and Dutton (2021) as examples for corals used as sea level indicators and Antonioli et al. (2015), Bini et al. (2014)) for studies using tidal notches.*

Line 50 use 'providing 'rather than 'suggesting 'maximum elevations
*A: Replaced.*

Line 55 include S+M 1972.. and maybe Dill, R.F., 1977, The blue holes, geologically significant submerged sinkholes and caves off British Honduras and Andros, Bahama Islands: 3rd International Coral Reef Symposium, Proceedings, v. 2, p. 237–242. It was recognised 50 years ago that such deposits would be useful see Benjamin (1970) in National Geographic

*A: Based on your recommendation, we included the papers of Benjamin (1970) and Spalding and Mathews (1972).*

Line 82. Age of carbonate material below hiatus is a _maximum_ age for timing of submergence (assuming no post-depositional alteration of exposed surfaces)
*A: Thank you for bringing this typo to our attention; we have changed to (line 115): "Dating the carbonate layer immediately above each of these hiatuses provides a minimum estimate of when the cave became air-filled again constraining the minimum age for the sea-level fall. The carbonate layer below a hiatus indicates the maximum age, assuming no post-depositional alteration of the exposed surfaces, for when this location in the cave was air-filled and the sea level was clearly below the speleothem elevation."*

Line 83. Avoid the 'dipstick 'analogy.. dipsticks indicate level, not binary status.
*A: We agree, we deleted this analogy and we rephrased to: "Hence, they stop growing when sea levels are above their elevation and deposition commences when sea level falls below them".*

Line 84.. choose alternative to roughly.. constraints are robust and accurate (elevation), but there may be lag between the constraint and changing water levels.
*A: This has been changed in the text.*

Line 87 on. Use the term minimum estimate. I would challenge that initiation of growth can be rapid after emerence, but difficult to predict. The likelihood/duration of lagged response has yet to be established. Rough estimate is overstating things. Another advantage, not mentioned in the work is that further growth over the earliest calcite, post emergence, is 'protected 'by subsequent growth. This is not the case for ages prior to hiatuses, which are susceptible to diagenetic alteration.
*A: We adjusted the text accordingly and we also included: "It is worth noting that the earliest layers deposited above the hiatuses are protected by further carbonate precipitation, whereas those below the hiatuses are susceptible to diagenetic alteration or even dissolution."*

Line 94.. they do not document the "moment".. they can constrain the maximum and minimum age of sea level change at this depth.
*A: Agree, we have changed the text to: "Hence, while these hiatuses can be chronologically well constrained, the vadose speleothems can only provide the maximum and minimum age when a particular part of the cave became flooded or air-filled…".*

Line 95. Time will tell.. speleothems have the potential to document the rise and fall of sea levels in stage 5. Maye the that ideal sites have not been identified yet.
*A: We agree, vadose speleothems have indeed the potential to provide more accurate results if carefully and comprehensively analyzed.*

Line 96. Choose alternative to 'difficult'. The data are unambiguous and easy to interpret, once it is established that they are constraints and NOT sea level index points.

*A: We updated the sentence to: "Therefore, the growth of vadose speleothems is a sea-level terrestrial limiting point and thus, its relationship with the sea level position must be interpreted correspondingly."*

Fig. 1. A useful figure, but needs refinement. e.g. constraints on sea level in (b).. label vadose calcite and hiatus/biogenic overgrowths. (c).. grey background, poor reproduction and use of T1 to T3 will have reader question why the internal morphology with changing MSL is not illustrated. POS should start earlier than T1 and at lower elevations than illustrated.

*A: The reviewer is correct on that POS will begin their growth slightly before T1, however, if the rate of sea-level rise is high, the amount of carbonate encrusting the pre-existing speleothem will be insignificant. Instead, when sea level rise is very slow or remains stable for at least 200-300 years, the POS develop their unique morphology. This is the reason why we showed only a fully developed POS at time T1 (within the tide range) and its subsequent evolution to T3. We addressed your other comments and revised the figure accordingly.*

[Figure]

*Figure 1 a) Composite diagram showing how SVS (left) and POS (right) in littoral caves act as sea-level indicators. b) Conceptual model showing that: b1) Growth of vadose speleothem (vs) indicates times when sea level (sl) was lower than their elevation; b2) Biologic encrustation (be) suggests sea level higher than the SVS elevation; b3) SVS resumes its deposition when sea level fell below its elevation; b4) Sea-level rise causes the deposition of a second biologic encrustation (growth hiatus). c) SVS from Argentarola Cave (Italy; photo courtesy F. Antonioli). d)*

*Conceptual model showing how POS form; as long as sea level remains at the same elevation (T1-T3), POS will precipitate within the tidal range and will continue to grow until the sea level drops below the speleothem (T4). e) POS in Cala Varques Cave, Mallorca. f) Mushrooms-shaped POS in Santa Catalina Cave, Cuba (photo courtesy B. P. Onac).*

Line 116.. shorten to 'Cave deposits have received little or no attention in prior compilations'.
*A: We updated the sentence in question in accordance with your suggestion.*

⸺

Section 2
Line 146 'varying 'quality.. avoid. I'd accept 'variable'.. but you have not discussed criteria thus far. And on this note, it would be worth illustrating the range in RSL data quality (1 to 5, compiler defined) For this section, MIS stage 5 boundaries might be determined by material that falls outside MIS 5 (e.g. Bahamas data in Li et al 1989; Richards et al, 1994; Lundberg and Ford 1994).
*A: We adjusted the text accordingly and also included the following information: "All POS records have precisely measured elevation, a narrow indicative range, and a sub-metric RSL uncertainty, hence they are excellent sea level index points. A guide for SVS records' evaluation as terrestrial limiting points can be found in Table 1 of WALIS 'official documentation (https://walis-help.readthedocs.io/en/latest/Relative%20Sea%20Level/).*

Table 1. Sites For 'not mentioned '(Moseley et al, 2013) – numerous caves in Quintana Roo, or similar. Additional information on Bahamas samples/sites. Richards et al., 1994. AN samples are from Stargate, Andros; GB samples Sagittarius Cave, Grand Bahama.
*A: We updated the table accordingly.*

| Location | Cave name | Type of cave | Reference |
|---|---|---|---|
| Mallorca | Cova de Cala Varques A | POS | Dorale et al., 2010; Polyak et al., 2018 |
| | Cova de Cala Varques B | | Polyak et al., 2018 |
| | Cova del Dimoni | | |
| | Cova de Cala Falcó | | |
| | Cova des Pas de Vallgornera | | |
| | Cova Genovesa | | |
| | Cova de s'Ònix | | |
| | Coves del Pirata | | |
| | Cova des Serral | | |
| | Coves del Drac | | |
| | Cova de sa Tortuga | | |
| Sardinia | Grotta di Nettuno | POS, SVS | Tuccimei et al., 2007 |
| Cuba | Santa Catalina Cave | POS | De Waele et al., 2017; 2018 |
| Bermuda | Government Quarry Cave | SVS | Harmon et al., 1978; 1981 |
| | Bierman Quarry Cave | | |
| | Crystal Cave | | |
| | Wilkinson Quarry Cave | | Wainer et al., 2017 |
| Yucatan Peninsula | Ccaves in Quintana Roo | SVS | Moseley et al., 2013 |
| Krk Island, Croatia | U Vode Pit | SVS | Surić et al., 2009 |
| Andros Island, Bahamas | Blue Hole in South Bight | SVS | Gascoyne et al., 1979 |
| | Stargate | SVS | Richards et al., 1994 |
| Grand Bahama, Bahamas | Lucayan Caverns | SVS | Lundberg and Ford, 1994 / Richards et al., 1994 |
| | Sagittarius Cave | SVS | Richards et al., 1994 |

Line 160. Barometric altimeter (or diver depth gauge) adjusted for density profile, where possible with information about the vertical density profile associated with fresh- water, brackish and saline zones. This is declared in database but referred to in discussed in text.

*A: We added the following text at the end of the Elevation measurements and their uncertainties section: "We recommend that the elevations of submerged samples to be corrected for density variation in the water column whenever salinity profiles are available and that the vertical density profile associated with freshwater, brackish, and saline zones to be included with the depth information when using barometric altimeter (or diver depth gauge) in future work."*

Line 177. Sample ID – samples may have been referred to in a number of papers. Declare best endeavour to find the first occurrence for sample ID might be useful phrase here.

*A: Reviewer's point is valid. We added the following text:*

*"Since samples once collected may end up having several IDs (collection, dating lab, etc.), it is recommended that whenever included in a database, authors should always use the ID associated with its first description and when the lab ID is different, this should also be added."*

Line 180. Reported ID - this might be sample ID or lab code. Database may need expanding to reflect this. For many geochronology laboratories, the analysis has separate codes to that provided by the group responsible for sub-sampling (e.g. Moseley et al 2013).

*A: We understand your concern. In order to avoid any confusion and to include all details, our best attempt for cases like Moseley et al. (2013) was to refer to the Reported ID as the Sample ID in the paper but including the lab code as well. To clarify this for our readers we added in the manuscript the following sentence:*

*"Finally, the Reported ID is the published sample identifier or Laboratory ID offered by the authors in the original paper. Since samples once collected may end up having several IDs (collection, dating lab, etc.), it is recommended that whenever included in a database, authors should always use the ID associated with its first description and when the lab ID is different, this should also be added. We note that for samples that have different Laboratory ID than the Sample identifier, i.e., Moseley et al. (2013), both are included in our Reported ID."*

Line 192. Please allow for possibility of using broken speleothems where possible, to preserve the aesthetic quality of caves. For many cases, the original location of a broken sample can be established.

*A: We addressed this point above in your "General comments" section.*

Line 194. Mineralogy does not affect the reliability.. it may dictate the susceptibility to alteration, but equally important is the geochemical setting. Low magnesium calcite is a relatively stable form of calcium carbonate in fresh and/or saline water, but mixing zone corrosion can cause serious dissolution. Encourage use of petrographic investigation. Also, marine borings, encrustations etc can dramatically alter the isotopic signal

*A: In response to your comment, we revised the first part of this paragraph that now reads (lines 255-266):*

*"The geochemical setting and sample mineralogy may dictate the susceptibility to alteration. For example, samples that show conversion of aragonite to calcite or calcite recrystallization, could have been subjected to uranium loss, which is an important factor that impacts U-Th ages (Lachniet et al., 2012; Bajo et al., 2016). To allow recognition of diagenetic fabrics, XRD screening is desirable for dating purposes (aragonite is preferred vs calcite). In order to make the best selection of samples, we encourage the use of petrographic investigation as well. Thin sections can help to better identify the speleothem layers just below and above the hiatus for dating purpose. Also, they reveal the internal structure and hence, areas affected by recrystallization can be avoided for sampling. However, only some of the studies compiled here report the mineral assemblage of the samples by X-ray diffraction (Surić et al., 2009; De Waele et al., 2017; 2018). De Waele et al. (2018), for example, complemented the screening method with petrographic investigations (thin sections) and imaging using scanning electron microscopy. We do not exclude the possibility that screening was performed in the other publications, but it has been not reported. For future studies, we strongly recommend including information on mineral assemblage, as well as diagenetic and crystalline descriptions."*

Line 204 Dutton et al (2017) "is very applicable... for speleothems aswell". It was written to apply to all U-Th geochronology. Please reword.
*A: We addressed this point and avoided redundancy.*

Line 209 use 'alpha spectrometry', or 'alpha counting'. 'Alpha detector 'is use in database, which is usually reserved for more basic instruments.
*A: We used "alpha counting".*

Line 211 use measured isotopic 'ratios and concentrations', not 'characteristics'.
*A: Replaced.*

Line 212. Use 'analyses of well-characterised internationally-recognised standards or certified reference materials are used to demonstrate the reliability of results'.
*A: This information has been now included in the text.*

Line 213 'reduced 'age uncertainties – not 'lowered'
*A: This has been replaced in the revised manuscript.*

Line 214 delete 'progressively'
*A: Deleted.*

Line 217 insert 'reported alpha spectrometry . . .or TIMS _results_'
*A: Inserted.*

Line 220 insert used: initial ratios and the decay constants used
*A: Inserted.*

Line 221 delete the effect of detrital thorium concentration. NB it is not only detrital Th one needs to consider, but also what some would call hydrogenous Th. Use of a threshold value of 300 (Helstrom, 2006) for 230Th/232Th is arbitrary and depends on the ratio used for initial Th.

*A: Reviewer #2 raised the concern that detrital correction in speleothem dating is critical and suggested that we should include the different procedures used by laboratories to apply this correction. We tried to address both reviewers' suggestions on this point and the text now reads as follows (lines 285-295):*

*"The correction for the initial non-radiogenic sources (i.e., hydrogenous, colloidal, and carbonate or other detrital components; Richards et al., 2012) of $^{230}$Th incorporated at the time of speleothem deposition is extremely important for age calculation and is sensitive for samples that contain very little uranium or an abundance of detrital thorium. The $^{230}$Th/$^{232}$Th activity ratio of 0.825 with an arbitrarily assigned uncertainty of 50% found in the mean bulk Earth or upper continental crustal has been commonly assumed for initial $^{230}$Th corrections. However, several studies have shown that this value may not cover all situations. Therefore, laboratories apply different corrections for the non-radiogenic detrital $^{230}$Th fraction through either direct measurement of sediments associated with speleothems (Hoffmann et al., 2018) or computed isochron methods and stratigraphical constraints (Hellstrom, 2006; Richards et al., 2012). Most POS included in this database fulfill the criterion suggested by Hellstrom (2006) that samples with ratios of $^{230}$Th/$^{232}$Th higher than 300 are considered clean, with very few samples having lower values (Tuccimei et al., 2007). However, the use of this threshold value is arbitrary and depends on the ratio used for initial Th."*

Line 230 on. This text needs to be improved. I think it is fair to accept that if ages are not presented as BP (or AD1950) or yr b2k, you can assume that dates are calculated with respect to date of analysis.

*A: We updated the sentence in question in accordance with your suggestion.*

Line 232 declare ±100 years (2 sigma) for age uncertainties for material of last interglacial age and add typical U concentration and declare insignificant 230Th initial

*A: However, with the improvements made using MC-ICP MS (Cheng et al., 2013), ages on good quality samples (i.e., aragonite mineralogy, high U content, insignificant $^{230}$Th correction) of last interglacial are now possible to uncertainties of ± 100 years (2σ), making how the age is reported more important (i.e., BP)."*

3. Discussions
Line 236. correlated to the _same_ (insert?)

*A: We completed this sentence which now reads: "The elevation of a sea-level indicator is not always coincident with the position of relative sea level (RSL) at the time of its formation, but rather is correlated to it by a quantifiable relationship."*

Line 250 replace 'clear' with 'unambiguous'

*A: Replaced.*

Line 255 delete 'absolute', also declare early in text that all age uncertainties are quoted at 2 sigma.

*A: Done. We also replaced "errors" with "uncertainties". We added at the end of Section 2.6: "Except for Gascoyne et al. (1984, 1979) in which uncertainties are reported as 1σ, Harmon et al. (1978) as standard error, and not mentioned in Harmon et al. (1981), all ages are reported with 2σ absolute uncertainties. As Dutton et al. (2017) suggested, we strongly recommend the use of the term "uncertainty" which is more appropriate to use in this context than the term error."*

Figure 3. You declare all ages are corrected for detrital Th. Begs the question – in original paper?, using updated decay constants? Please add more information.
*A: We agree that further explanation is needed here. We updated the figure and its caption which now reads:*

*"Paleo RSL position recorded by POS. Except for Tuccimei et al. (2007), all U-Th ages are corrected for detrital Th as per the original publications. No other further corrections have been applied."*

[Figure]

Line 281. Calcite growth above and below (or before and after) a growth hiatus (not bottom and top).
*A: Replaced.*

Line 283. Again, avoid use of 'roughly'
*A: Done.*

Line 288 from not form; deciphering the timing of stillstands – if that is what you mean
*A: Yes, thank you for these corrections.*

Line 290.. be consistent you use mapsl elsewhere and m here. Is there a difference?
*A: No, there is no difference. We adjusted the text accordingly.*

Para L300 on. Andros – in addition to the data for Gascoyne et al (1979), there are data in Gascoyne (1984), Richards et al (1994), Smart et al (1998) - see references above. Mostly alpha-spectrometric or TIMS U-Th ages.

*A: We included these references in the database and the ages that are within the interval targeted here are discussed in the text.*

Figure 4 does not include dates from Andros (Gascoyne, 1984; Richards et al 1994) or Grand Bahama (Richards et al, 1994; Smart et al, 1998). Not necessary, but there is a suggestion that data relating to MIS 5 in broadest sense is included. Also.. plotting U-Th ages alone only goes so far. The duration of continuous growth is required (see earlier comment). This figure does not illustrate this. It is not necessary, but the point needs to be made that these are ages only and _not_ growth periods. Awkward placement of legend, it distracts from the data.

*A: As suggested, we updated Figure 4. We extended the age range from 150-70 ka and we included dates from the indicated references that overlap with this time interval. The figure caption now reads: "Figure 4. SVS (terrestrial limiting points) sample elevations and their U-series ages indicating RSL below (down-pointing triangles) or above (up-pointing triangles) them. Note that these are ages only and not growth periods. None of the data are corrected for GIA or long-term deformation."*

[Figure]

*It was not clear to us if you suggested to add horizontal bars to highlight the growth periods of some of the speleothems instead of their ages only. If this is your request, we can update the figure in the revised manuscript.*

Line 365. The Bahama archipelago (includes the Bahamas and Turks and Caicos).
It is worth noting the work conducted by the Miami group to assess the subsidence of the Bahamas, in part because of loading through periodic carbonate production on the platform. e.g. McNeil, D.F.; Ginsburg, R.N.; Chang, S.-B.R., and Kirshvink, J.L., 1988. Magnetostratigraphic dating of shallow water carbonates from San Salvador, Bahamas. Geology, 16(1), 8–12.

*A: Thank you for your suggestion. We cited instead a more recent paper of McNeil (2005) and we included in the manuscript (line 454):*
*"In addition to the GIA corrections, the long-term subsidence of the platform needs to be also considered when assessing the LIG sea-level record. Subsurface drill core data from shallow water stacked facies indicate that the Bahamas is subsiding at rates of several meters per 100 kyr (McNeill, 2005)."*

In addition.. it would be worth acknowledging the additional constraints on sea levels that are provided by the flank-margin caves that host such speleothem deposits in some places. Perhaps at the same time as notches are mentioned (e.g. papers by Mylroie et al., 2020; Carew and Mylroie, 1990; Mylroie and Carew, 1995).
*A: We thank you for suggesting these additional references. We included flank margin caves as LIG sea level indicators in the introduction and the text has been updated to: "Other indicators such as erosional notches (Bini et al., 2014; Antonioli et al., 2015;) or flank margin caves (Carew and Mylroie, 1990; Mylroie et al., 2020) pinpoint sea level, but lack tight age control."*

From journal guidelines: Ma and Myr (also Ga, ka; Gyr, kyr): "Ma" stands for "mega- annum" and literally means millions of years ago, thus referring to a specific time/date in the past as measured from now. In contrast, "Myr" stands for millions of years and is used in reference to duration (CSE, p. 398; North American commission on stratigraphic nomenclature).
*A: We updated the text accordingly.*

The data compilation/spreadsheet (speleothems - last interglacial from WALIS):
It would be useful to refer to the site https://walis-help.readthedocs.io/en/latest/ (or acceptable zenodo URL) earlier in body of text. A lot of critical relevant material can be found here.
File name needs to be more specific with date, perhaps? Or is this the standard file name for a query download? I note that other database compilations have name of first author compiler and version/ date.
*A: We agree with your suggestion and have now referred the readers to the official documentation of the WALIS database. Also, thank you for suggesting a more specific name for the file. We will upload a new version of the database and will update the file name as well.*

Tuccimei et al. URL link is broken
*A: The link was updated and now opens the file on the journal's page at:*
*http://www.museucienciesnaturals.org/userFiles/File/Publicaciones/articles_cientifics/08tuccimei_cuerda.pdf*

For data on vertical movements – what qualifies here? Needs guidance.
*A: This section includes vertical land movement due to tectonics. Data should be compiled only if independent vertical land motions are available for the site (https://walis-help.readthedocs.io/en/latest/RSL_data.html).*

Please comment on the "Quality of age information", or signpost where this information can be found for speleothems in particular.
*A: Following your recommendation, we refer the reader to evaluation guide on the Age information quality and included in the manuscript the following text:*

*"We refer the reader to the guide on the evaluation of the ages' quality which can be found in WALIS' official documentation at: https://walis-help.readthedocs.io/en/latest/RSL_data.html.*

A component that needs to be considered is the definition of continuous growth. The age of growth initiation and cessation is dependent on sample position and growth rate. This caveat needs to be included in the paper. Would the database be expanded to include such information?
*A: We acknowledge the importance of this caveat and addressed the concern related to the continuous growth earlier in our response above. We included this caveat in the revised manuscript (lines 137-145). We agree that this would be a useful addition to the database.*

Reported age – please define (is this uncorrected for initial 230Th, for example). And what does corrected reported age mean? There is information at the WALIS website, but this should be included in the text for this paper.
*A: This concern has been addressed earlier in the General comments section.*

Please reflect on the reporting of symmetrical U-Th age uncertainties for last interglacial ages. delta234U (declare that these are per mil values).
*A: For consistency with the format of WALIS database we report all U-Th ages with symmetrical uncertainties. To do so we calculated an average of the asymmetrical uncertainties given by Wainer et al. (2017). This is noted in the "Comments on the age determination" column: "The original authors report $^{232}Th/^{230}Th$ values but for consistency we calculated and included them in the database as $^{230}Th/^{232}Th$. Also, the authors report asymmetric uncertainties for the ages and for the initial $^{234}U/^{238}U$, however, we show here an average of those values and refer those interested in the asymmetric values to the Suppl. Table 1 of the original paper".*
*We also added in the per mil measure unit in columns CH and CI.*

There is an assumption that [U] = [238U].. e.g Dorale et al (2010) quote [U] ppm, but spreadsheet is [238U]*. Please comment. *Only 0.73% difference, but outside uncertainty of typical measurement.
*A: We checked with the authors and they confirmed that they measured $^{238}U$.*

In database.. please distinguish between activity ratios and abundance ratios. Generally OK, but see heading for 230Th/232Th.. this is activity ratio.
*A: We assume that you may be referring to the column of $^{230}Th/^{232}Th$ initial which indeed was not mentioned that it was activity ratio – we have now updated the column head to [$^{230}Th/^{232}Th$ initial]ACT.*

Report latitude and longitude to full precision declared in paper ie. 87.00 rather than 87 to avoid confusion (see Moseley et al, 2013).
*A: This has been updated in the database spreadsheet. We would like to note that the coordinates are converted to decimal degrees to be consistent with the entire WALIS database.*

Explain the difference between coordinates and reported coordinates (latter not complete, see Cova del Dimoni, Mallorca).
*A: To clarify this, we added in the revised manuscript: "Except for the SVS from Yucatan Peninsula reported by Moseley et al. (2013) and from Bermuda (Wainer et al., 2017), studies do*

*not report the exact cave location from where samples were collected. Hence, the latitude and longitude for these indicators were determined using Google Earth to match locations from publication maps and noted accordingly."*

*We also clarified this difference by adding a note in the "Comments on geographic coordinates" column of the database spreadsheet, which reads: "estimated using Google Earth" for all the coordinates that have not been reported by the authors.*

Additional references referred to in text above
Mylroie, J.E. and Carew, J.L. (1990) The flank margin model for dissolution cave development in carbonate platforms. Earth Surface Processes and Landforms, 15(5), 413–424.
Carew, J.L. and Mylroie, J.E. (1995). Quaternary tectonic stability of the Bahamian Archipelago: Evidence from fossil coral reefs and flank margin caves. Quaternary Science Reviews, 14(2), 144–153.
Mylroie, J, Lace, M., Albury, N and Mylroie, J. (2020) Flank Margin Caves and the Position of Mid- to Late Pleistocene Sea Level in the Bahamas. Journal of Coastal Research, 36(2), 249-260.
Gascoyne M. (1984) Uranium-series ages of speleothems from Bahamian blue holes and their significance. Cave Science, 11(1) 45-49.
Smart PL, Richards DA, Edwards RL. (1998) Uranium-series ages of speleothems from South Andros, Bahamas: Implications for Quaternary sea-level history and palaeocli- mate. Cave and Karst Science 25(2), 67-74.

*A: We included all these references in the manuscript as well as all the others referred to in the text above.*

*We would like to add that based on Rev. no 2 and your recommendations, we consider including in the manuscript the following section with the necessary information that we strongly encourage researchers publishing new sea-level studies based on speleothems to report.*

*"To build a more valuable dataset that will have more longevity and use within the discipline, we strongly encourage researchers publishing new sea-level studies based on speleothems to include the following information:*

- *Sea-level indicator and its relationship to sea level: i) site location (latitude and longitude of the cave); ii) the elevation of the sea-level indicator, the instrument type used and its precision, and the error associated with the elevation measurement (when using barometric altimeter or diver depth gauge for submerged samples, the elevations should be adjusted for density profile); iii) the sea level datum to which the elevations are referred and how the indicative meaning has been quantified.*
- *Screening results: XRD, petrography, and polarizing/scanning electron imaging, including information on mineral assemblage, as well as diagenetic and crystallization descriptions (e.g., fabric).*
- *U-series data: In order to collectively improve the utility of U-series data we encourage researchers publishing new sea-level studies based on speleothems to follow the recommendations suggested by Dutton et al. (2017) in reporting their data. These authors specify the required data to enable calculation and, if needed, re-calculation of the same ages using different parameters and also, to facilitate the interpretation in the*

*context of other studies. The checklist of minimum data to report includes: uncertainties for all parameters, state whether uncertainties on ages include decay constant uncertainties; Names, descriptions, and reference values of reference materials; Decay constants; Isotopes in spike and method of spike calibration; Method of calibration for all activity or atom ratios reported; Activity or atom ratios for $^{230}Th/^{238}U$ (or $^{230}Th/^{234}U$) and $^{234}U/^{238}U$; $^{230}Th/^{232}Th$ activity or atom ratio; Details of procedures and values used to interpret ages using isochrons or other models; Date of analysis or reference age (e.g., BP, b2k, etc.). These recommendations will increase the usefulness of this type of analytical results in the U-series geochronology community (Dutton et al., 2017). We also recommend reporting continuous growth rate which allows to better define the onset and cessation of deposition for either POS or SVS samples."*